# HB-EGF activates EGFR to induce reactive neural stem cells in the mouse hippocampus after seizures

Oier Pastor-Alonso[1], Irene Durá[1], Sara Bernardo-Castro[1], Emilio Varea[2] , Teresa Muro-García[1] , Soraya Martín-Suárez[1], Juan Manuel Encinas-Pérez[1,3,4] , Jose Ramon Pineda[1,5]

Hippocampal seizures mimicking mesial temporal lobe epilepsy cause a profound disruption of the adult neurogenic niche in mice. Seizures provoke neural stem cells to switch to a reactive phenotype (reactive neural stem cells, React-NSCs) characterized by multibranched hypertrophic morphology, massive activation to enter mitosis, symmetric division, and final differentiation into reactive astrocytes. As a result, neurogenesis is chronically impaired. Here, using a mouse model of mesial temporal lobe epilepsy, we show that the epidermal growth factor receptor (EGFR) signaling pathway is key for the induction of React-NSCs and that its inhibition exerts a beneficial effect on the neurogenic niche. We show that during the initial days after the induction of seizures by a single intrahippocampal injection of kainic acid, a strong release of zinc and heparin-binding epidermal growth factor, both activators of the EGFR signaling pathway in neural stem cells, is produced. Administration of the EGFR inhibitor gefitinib, a chemotherapeutic in clinical phase IV, prevents the induction of React-NSCs and preserves neurogenesis.

## Introduction

Mesial temporal lobe epilepsy (MTLE) is the most common form of epilepsy in adults (Falconer et al, 1964; Margerison & Corsellis, 1966). A portion of patients develop MTLE with hippocampal sclerosis (MTLE-HS), consisting of unilateral hippocampal atrophy, neuronal death, reactive gliosis, and granule cell dispersion (GCD). MTLE-HS is associated with drug resistance for anti-seizure medications, often leading to unilateral amygdalohippocampectomy (Crespel et al, 2005; Kwan & Sperling, 2009; Tatum, 2012). In the dentate gyrus (DG) of the hippocampus of most mammals, possibly including humans (Eriksson et al, 1998; Spalding et al, 2013; Llorens-Bobadilla et al, 2015), a population of neural stem cells (NSCs) continues to generate new neurons postnatally and throughout adulthood (Encinas & Enikolopov, 2008; Spalding et al, 2013). In human MTLE-HS and in experimental models based on the intrahippocampal injection of kainic acid (KA), a reduction in or the absence of markers of neurogenesis and cell proliferation in the hippocampal neurogenic niche has been reported (Heinrich et al, 2006; Ledergerber et al, 2006; Nitta et al, 2008). In other models of MTLE (based on pilocarpine administration or the electrical induction of seizures), neurogenesis was found to be transiently increased (Parent et al, 1997; Madsen et al, 2000). In parallel to alterations in neurogenesis, seizures trigger astrogliogenesis in the DG in both mice and humans (Ammothumkandy et al, 2022). We have recently characterized, using mouse models of MTLE-HS, how seizures induce the transformation of hippocampal NSCs into reactive NSCs (React-NSCs) that contribute to HS through the direct generation of reactive astrocytes at the expense of neurogenesis (Sierra et al, 2015; Muro-García et al, 2019; Valcárcel-Martín et al, 2020).

The mechanisms controlling the differentiation of NSCs into React-NSCs in the intrahippocampal KA mouse model have not yet been investigated and could be of therapeutic use, as preserving the population of hippocampal NSCs in MTLE-HS could (1) preserve neurogenesis and its normal functions, (2) allow for the endogenous restoration of the GCL neurons lost to excitotoxicity, and (3) reduce reactive gliosis. Herein, we studied the involvement of the epidermal growth factor receptor (EGFR) in the induction of React-NSCs and whether it could be a potential target to preserve them after seizures. EGFR has been directly reported to be present in proliferative embryonic neural precursors (Ciccolini et al, 2005) and in activated hippocampal adult NSCs and amplifying neural progenitors (ANPs) (Jhaveri et al, 2015; Walker et al, 2016). Although under normal conditions brain astroglia weakly express EGFR, it plays a role in the proliferation and differentiation of astrocytes (Simpson et al, 1982). In addition, a dramatic increase in EGFR has been observed after brain injury or focal ischemia, especially in reactive astrocytes and microglia (Nieto-Sampedro et al, 1988; Planas et al, 1998). Increased levels of EGFR promote the ability of

[1]Laboratory of Neural Stem Cells and Neurogenesis, Achucarro Basque Center for Neuroscience, Bizkaia, Spain  [2]Faculty of Biology, University of Valencia, Valencia, Spain  [3]Ikerbasque, The Basque Foundation for Science, Bizkaia, Spain  [4]Department of Neurosciences, University of the Basque Country (UPV/EHU), Bizkaia, Spain  [5]Signaling Lab, Department of Cell Biology and Histology, Faculty of Medicine and Nursing, University of the Basque Country (UPV/EHU), Bizkaia, Spain

Correspondence: jr.pineda@achucarro.org; jm.encinas@achucarro.org
Oier Pastor-Alonso's present address is Department of Neurology, University of California San Francisco, San Francisco, CA, USA

adult NSCs to differentiate into astrocytes (Burrows et al, 1997), and asymmetric distribution of EGFR in late cortical progenitors promotes the astrocyte lineage in cells expressing high levels of EGFR (Sun et al, 2005). Moreover, it has been recently postulated that the genetic deletion of EGFR affects the self-renewal capacity of neural stem and progenitor cells (NSPCs) from the subventricular zone (SVZ), promoting the differentiation of the progeny toward astrocytes (Robson et al, 2018). However, mice lacking EGFR show the reduced numbers of cortical astrocytes and increased hypersensitivity to epileptic seizures induced by intraperitoneal KA administration (Sibilia et al, 2007; Robson et al, 2018). Furthermore, our previous data on functional gene analysis from the hippocampus of mice subjected to MTLE-HS showed, among others, an early upregulation of the ErbB transduction pathway (Sierra et al, 2015). The ErbB protein family contains four structurally related receptor tyrosine kinases, including the EGFR. Both EGFR and fibroblastic growth factor receptor (FGFR) signaling pathways have been demonstrated to be highly mitogenic for astrocytes (Simpson et al, 1982; Lewis et al, 1992). Also, FGFR expression increased in the cortex and hippocampus of rats injected intraperitoneally with KA (Van Der Wal et al, 1994), but no information exists regarding the EGFR.

## Results

### Seizures cause an early overexpression and activation of the EGFR signaling pathway in the hippocampus

We first sought to evaluate the expression of both EGFR and FGFR in a standardized model of MTLE-HS consisting of a single administration of 1 nmol of KA in the DG of C57BL/6 mice (Bouilleret et al, 1999; Sierra et al, 2015). The hippocampus was collected at early time points after the KA injection (1.5, 12, 24, and 72 h) (Fig 1A) and analyzed by the real-time quantitative polymerase chain reaction (qRT–PCR) (Fig 1B). A twofold-to-threefold significant increase in EGFR mRNA was found as early as 1.5 h and was maintained at 24 and 72 h post-KA (one-way ANOVA, $P = 0.028$, $P = 0.037$, and $P = 0.007$, respectively, compared with controls; Fig 1B). In order to measure the activation of the EGFR pathway, we determined the phosphorylation of the receptor (P-EGFR) on its tyrosine site 845 (Tyr845), which has been shown to be required for triggering DNA synthesis (Boerner et al, 2005). The Western blotting (WB) analysis revealed a progressive increase in P-Tyr845 (with respect to the normal EGFR) that became significant at 24 and 72 h post-KA ($3.116 \pm 0.698$ at 24 h and $8.540 \pm 0.357$ at 72 h compared with $1.000 \pm 0.385$ (control); one-way ANOVA, $P = 0.04$ and $P$ 0.001, respectively; Fig 1C and D). In addition, we measured the activation level, also by phosphorylation, of different downstream effectors of the EGFR signaling pathway in the hippocampal samples (Fig 1C). EGFR mediates the activation of Janus kinase/signal transducer and activator of transcription 3 (STAT3) (Ueno et al, 1997); AKT, involved in cell survival; and ERK1/2, which regulates cell stress and proliferation (Meloche & Pouysségur, 2007; Lill & Sever, 2012; Goffin & Zbuk, 2013). Our results showed that STAT3 was activated 12-fold compared with controls at 12 h post-KA injection and then decreased (Kruskal–Wallis test, $P = 0.026$; Fig 1E). AKT phosphorylation

increased gradually, becoming significant at 72 h (15-fold increase with respect to the control; Kruskal–Wallis test, $P = 0.043$; Fig 1F). Phosphorylation of ERK1/2 was more prominent, becoming statistically significant as early as 1.5 h post-KA (eightfold with respect to the control) and then remaining elevated afterward (10-fold to 20-fold with respect to the control; one-way ANOVA, $P = 0.023$, $P = 0.05$, $P = 0.05$, and $P$ 0.001, respectively, compared with controls; Fig 1G). In contrast, no differences were found in the amount of FGFR1 or FGFR2 mRNA (Fig S1A), or in FGFR1 protein (Fig S1B), in the hippocampus of MTLE-HS mice compared with control ones.

### EGFR expression increases after seizures in hippocampal NSPCs

EGFR plays a central role in promoting cell proliferation through the Ras/MAPK/ERK pathway (Schlessinger & Bar-Sagi, 1994; Cobb, 1999). Moreover, nuclear EGFR has been strongly correlated with high cell proliferation (Lin et al, 2001). Because a much higher rate of cell division is one of the characteristics of React-NSCs, we hypothesized that EGFR expression would be increased in MTLE-HS–induced React-NSCs. Thus, we determined EGFR expression in the hippocampal neurogenic niche of control and MTLE-HS mice. We used 2-mo-old transgenic Nestin-GFP (on C57BL/6 background) mice in which NSCs and React-NSCs can be readily visualized (Sierra et al, 2015; Muro-García et al, 2019). The mice were subjected to the intrahippocampal injection of KA (or saline for controls) and euthanized 24 h later. In controls, EGFR expression was sparse and tightly restricted to the neurogenic niche, specifically labeling a limited number of Nestin-GFP+ cells located in the subgranular zone (SGZ) of the DG (Fig 1H). In contrast, EGFR immunostaining had increased in the MTLE-HS mice and was noticeably prominent in more Nestin-GFP+ cells (Fig 1H). The quantification showed that EGFR expression (measured as pixels of positive immunostaining normalized for the respective regions) tended to be significantly increased in the GCL after KA (an increase of 400% versus control levels; $t$ test, $P = 0.057$; Fig 1I). The expression of the EGFR was also significantly increased in the hilus of the MTLE-HS mice (1,200% versus control levels; $t$ test, $P = 0.003$; Fig 1I). Specifically, colocalization of the EGFR with React-NSCs (MTLE-HS group) was increased by 450% compared to colocalization with control NSCs ($t$ test, $P = 0.031$; Fig 1I). We found no changes in FGFR1 expression by immunofluorescence in the hippocampal neurogenic niche (Fig S1C), confirming the result obtained by qRT–PCR and WB (Fig S1A and B).

### Pharmacological blockage of EGFR signaling reduces NSPC proliferation in vitro and in vivo

We next aimed to directly modulate EGFR signaling in NSPCs without the interference of the profusion of external niche signals present in vivo. We isolated and enriched hippocampal NSPCs in vitro (adaptation of Pineda et al [2013]) from 2-mo-old Nestin-GFP mice and first confirmed that 1 wk after primary culture and neurosphere generation, the EGFR was present in all cells (Fig S1D). However, NSPCs undergoing mitosis, as identified by their chromosome condensation typical of prophase and metaphase (confirmed by co-immunostaining for the mitotic marker Ki67), expressed significantly higher levels of EGFR, as quantified by

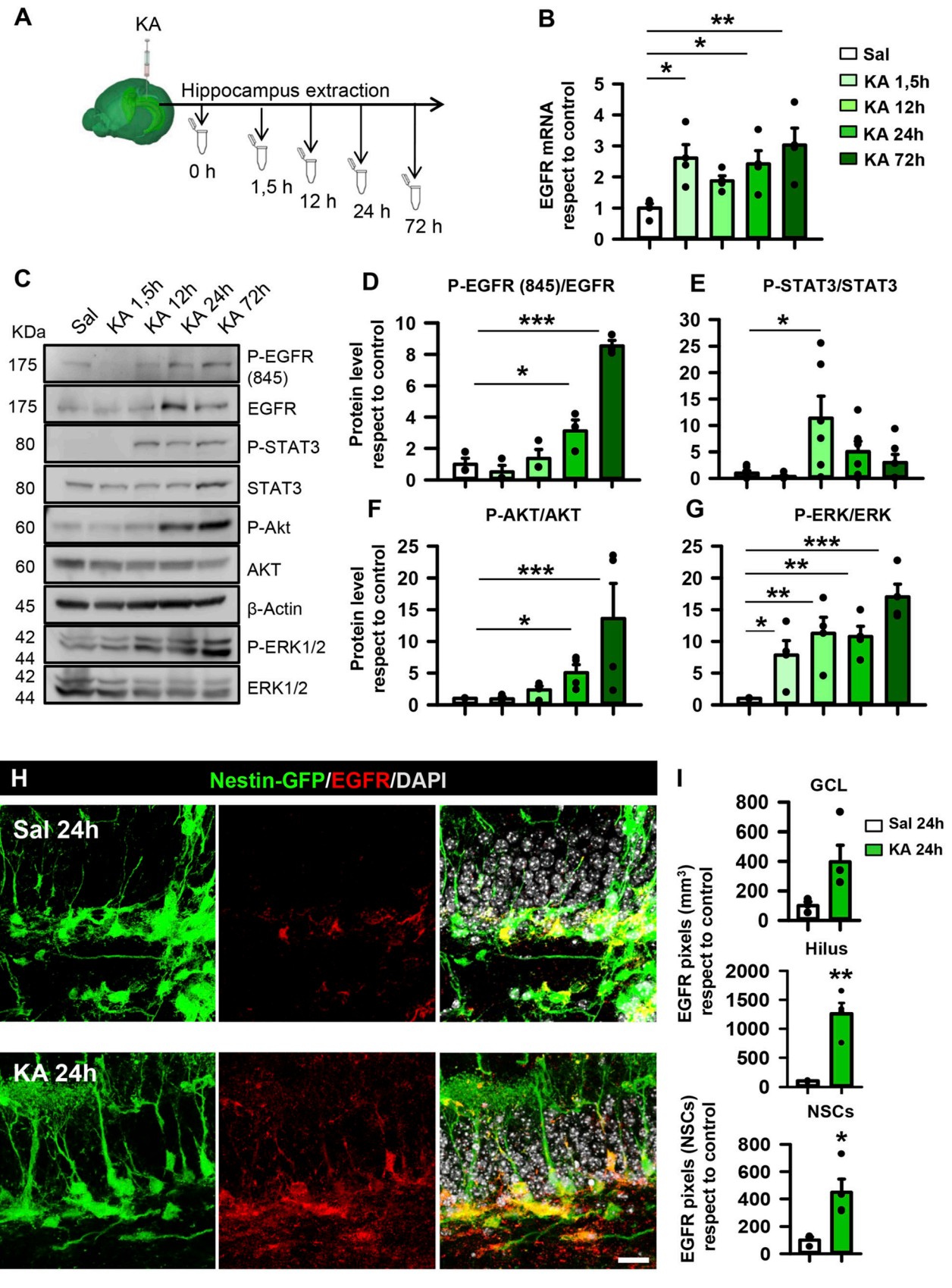

measuring signal intensity after immunostaining for EGFR (2.34 ± 0.18 versus 0.81 ± 0.05 pixel/$\mu m^2$; Mann–Whitney test, $P$ 0.001; Fig S2A and B). We then determined the effect of the pharmacological inhibition of EGFR signaling using the reversible inhibitor gefitinib, which disrupts EGFR kinase activity by reversibly binding within the ATP-binding pocket of the EGFR protein, efficiently inhibiting all tyrosine phosphorylation sites on EGFR (Pedersen et al, 2005) (Fig 2A and B). We checked the effect of EGFR inhibition on both EGFR Tyr845 and Tyr1068, located within the activation loop of the kinase domain of EGFR. WB analyses showed that the addition of gefitinib (2 $\mu M$) for 1 h to EGF-stimulated (20 ng/ml) cultured NSPCs efficiently reduced phosphorylation at Tyr845 by 81.65% and at Tyr1068 by 69.47% with respect to control cultures without changes in total EGFR (Mann–Whitney test, $P$ = 0.029 for Tyr845 and $t$ test, $P$ = 0.031 for Tyr1068 and $P$ = 0.686 for EGFR; Fig 2A and B). Moreover, the presence of gefitinib efficiently reduced phosphorylation of the downstream effector ERK1/2 by 75.92% with respect to controls ($t$ test, $P$ = 0.012; Fig 2A and B). To corroborate the effects on cell proliferation, we cultured NSPCs in the presence of 2 $\mu M$ gefitinib for 48 h and administered a pulse of 10 $\mu M$ 5-bromo-2'-deoxyuridine (BrdU) 1 h before fixation in order to identify dividing cells (Fig 2C). We then costained for DAPI, BrdU, and Nestin-GFP (Fig 2D). Our results showed a 63% reduction in BrdU-positive cells in the presence of gefitinib with respect to the control (Mann–Whitney test, $P$ 0.001; Fig 2E). We also tested the effect of inhibiting EGFR on NSPCs using afatinib, an irreversible inhibitor of EGFR. NSPCs that were cultured for 24 h with afatinib (2 $\mu M$) had reduced proliferation, with 20.27 ± 5.95% of them being positive for BrdU in controls compared with 8.91 ± 4.88% in afatinib-treated cells (Mann–Whitney test, $P$ = 0.046; Fig S2C–E). However, this effect is attributable to the massive cell death induced by afatinib, as indicated by the dramatic loss of BrdU+ cells and Nestin-GFP+ NSPCs (plus the rounded and vacuolated morphology of the remaining ones) when NSPCs were cultured with afatinib for 48 h (Mann–Whitney test, < 0.001). As a confirmation of the deleterious effect of afatinib, the co-injection of KA plus afatinib (70 $\mu M$) in the hippocampus of Nestin-GFP mice also provoked a massive loss of Nestin-GFP+ and BrdU+ cells in the DG (Fig S2H and I).

## Blocking EGFR signaling with gefitinib partially preserves hippocampal NSCs after MTLE-HS in vivo

We therefore chose to continue with gefitinib to study the effect of EGFR inhibition on the hippocampal neurogenic niche in the mouse model of MTLE-HS. As gefitinib does not cross the blood–brain barrier, we administered it intranasally (McKillop et al, 2004), a pathway that has also been used for anti-epileptic drug treatment (Barakat et al, 2006). Although it has been reported that a single dose of gefitinib can block cell proliferation for 72 h (Pedersen et al, 2005), we administered it twice a day starting right after KA injection to maximize its presence over the 3 d after the induction of MTLE-HS, when the peak of increased proliferation is observed (Sierra et al, 2015). The results showed a 57.25 ± 4.98% reduction in proliferating Ki67+ cells in the SGZ of the animals treated with gefitinib compared with the animals treated with the vehicle DMSO ($t$ test, $P$ = 0.002; Fig 3A and B). Likewise, the number of proliferating (Ki67+) NSCs and of ANPs dropped to 45.25 ± 5.92% and 32.02 ± 5.25% ($t$ test, $P$ = 0.005 and $P$ = 0.007; Fig 3A and B). NSCs were identified as being immunopositive for nestin and glial fibrillary acidic protein (GFAP) and with a radial apical process extending across the GCL from the SGZ to the molecular layer. ANPs are located in the SGZ, are nestin-positive but GFAP-negative, and bear few and short processes.

In order to check the potential preservation of NSCs by EGFR inhibition in MTLE-HS, we repeated the administration of gefitinib or DMSO in KA- or saline-injected Nestin-GFP mice but waited 14 d to euthanize the mice and analyze them once the transformation into React-NSCs was fully completed (Fig 3C and D).

As expected, we found severe depletion of DCX+ cells in the KA+DMSO mice (8,914.61 ± 2,537.97 DCX+ cells) compared with saline+DMSO and saline+gefitinib groups (39,888.21 ± 6,887.70 and 31,341.6 ± 5,465.51 DCX+ cells, respectively; one-way ANOVA, $P$ = 0.022 and $P$ = 0.09; Fig 3D and E). In the MTLE-HS mice treated with gefitinib, substantial recovery of DCX+ cells was found in the DG, rising to 37,641.67 ± 8,077.953 DCX+ cells (one-way ANOVA, $P$ = 0.030; Fig 3D and E). At the end of the treatment, we administered three pulses of BrdU intraperitoneally, separated by 3 h, to determine the density of newly generated cells produced once the inhibition of proliferation had ceased. We observed a rebound trend in the saline+gefitinib (16,330 ± 2,550 BrdU+ cells), saline+KA, and KA+gefitinib (12,950 ± 2,430 and 14,190 ± 5,930 cells) groups and found no statistically significant difference compared with the saline+DMSO group (4,940 ± 3,470 cells; one-way ANOVA, $P$ = 0.256; Fig S3C).

The reactive astrocyte-like morphology typical of seizure-induced React-NSCs (Sierra et al, 2015; Muro-García et al, 2019; Valcárcel-Martín et al, 2020) was reduced as NSCs conserved their radial morphology in the MTLE-HS mice treated with gefitinib. To quantify this effect, we analyzed Nestin-GFP+ cells by Sholl analysis and measured the total process length, cell volume, and

**Figure 1. Increased expression of EGFR in the hippocampal neurogenic niche and activation of the EGFR signaling pathway are early events in MTLE-HS.**
**(A)** Schematic time course of hippocampus dissection after intrahippocampal injection of KA (1 nmol). **(B)** Determination of EGFR mRNA expression by qRT–PCR showing an early increase after KA injection. **(C)** Determination of changes in normal and phosphorylated protein levels involved in EGFR signaling by WB of hippocampal samples. p-EGFR, p-STAT3, p-AKT, and p-ERK are increased. Data are representative of three biological replicates, with representative blots cropped and marked with solid lines. **(D)** Quantification of the ratio of phosphorylated versus non-phosphorylated forms of EGFR. **(E)** Quantification of the ratio of phosphorylated versus non-phosphorylated forms of STAT3. **(F)** Quantification of the ratio of phosphorylated versus non-phosphorylated forms of AKT. **(G)** Quantification of the ratio of phosphorylated versus non-phosphorylated forms of ERK after WB. Phosphorylation changes are compared with respect to the control after WB. **(H)** Confocal microscopy images after immunostaining for EGFR in the SGZ of the hippocampus of control (upper panel) and MTLE-HS (lower panel) Nestin-GFP showing increased expression in NSCs and ANPs early after intrahippocampal KA injection. Scale bar, 20 $\mu m$. **(I)** Quantification of EGFR immunofluorescence signal in the GCL, hilar region, and NSCs. Data information: for (B), one-way ANOVA, *$P$ < 0.05 and **$P$ < 0.01. Bars show the mean ± SEM. n = 3. Dots show individual data. For (D, G), one-way ANOVA and Kruskal–Wallis test in (E, F), *$P$ < 0.05, **$P$ < 0.01, and ***$P$ < 0.001. Bars show the mean ± SEM. Dots show individual data. n = 5. For (I), Mann–Whitney test for GCL and $t$ test for the hilar region and NSCs, *$P$ < 0.05 and **$P$ < 0.01. Bars show the mean ± SEM. Dots show individual data. n = 3.
Source data are available for this figure.

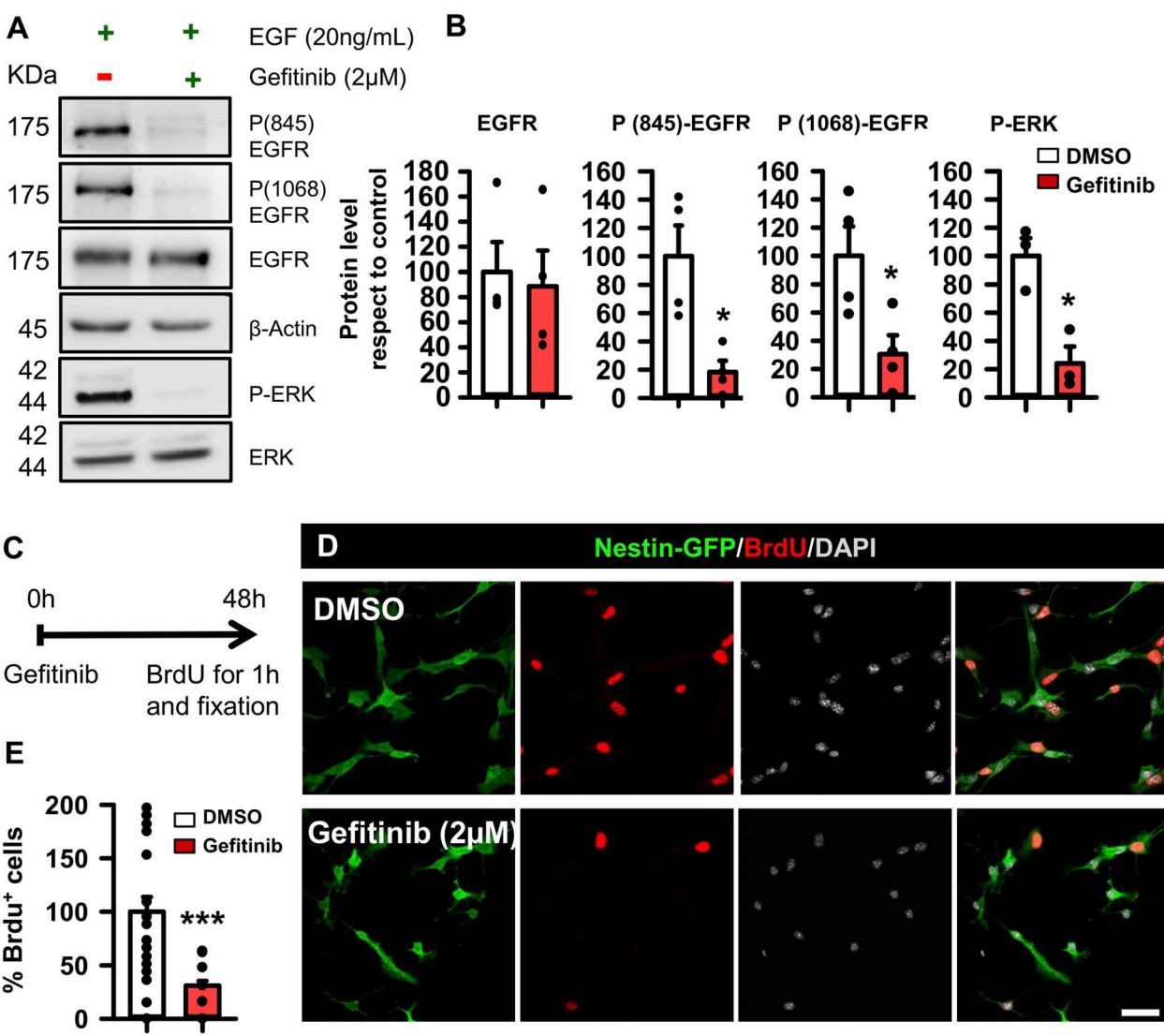

**Figure 2. Inhibition of EGFR signaling with gefitinib reduces proliferation of NSPCs.**
**(A)** EGFR of cultured hippocampal NSPCs stimulated with EGF was phosphorylated at Tyr845 and Tyr1068 residues as analyzed by WB. Its downstream effector ERK was also phosphorylated. Pretreatment with the EGFR inhibitor gefitinib blocked phosphorylation of the receptor and of ERK. **(B)** Quantification of the level of EGFR with respect to the loading control and quantification of the ratio of phosphorylated versus non-phosphorylated forms of EGFR Tyr845 and Tyr1068 and ERK (four independent replicates) showed the consistent inhibitory effect of gefitinib. **(C)** Paradigm of the strategy to evaluate the effect on cultured cell proliferation. Cells were cultured in the presence of EGF or gefitinib pretreatment plus EGF during 48 h. A 1-h pulse of BrdU 10 $\mu$M was given before fixation to label mitotic cells. **(D)** Representative immunofluorescence images of cultured NSPCs from Nestin-GFP mice showing the reduction of BrdU+ cells in gefitinib-treated cells. Scale bar, 10 $\mu$m. **(E)** Quantification of the number of BrdU+ cells expressed as the percentage with respect to total BrdU+ cells in the control condition showing the strong effect of gefitinib. Data information: for (B), Mann–Whitney test for Tyr845 and $t$ test for Tyr1068 and ERK, *$P < 0.05$. Bars show the mean ± SEM. Dots show individual data. n = 4. For (E), Mann–Whitney test, ***$P < 0.001$. Bars show the mean ± SEM. Dots show random fields of two pooled independent experiments.
Source data are available for this figure.

ramification as branch points (Fig 3F–H, respectively). In all three parameters, gefitinib significantly reduced the effect of KA. Notably, gefitinib also had an effect in saline-injected mice regarding the cell volume and the number of branch points. A pathological hallmark of MTLE-HS in the human hippocampus and animal models is GCD, which consists of the separation of granule cells resulting in increased length between the molecular layer and the hilus (Houser, 1990). We tested whether GCD was also reversed by the administration of gefitinib and found that indeed this was the case. A significant increase in GCD was found in the KA+DMSO

animals (1.012 ± 0.16 mm³; Kruskal–Wallis test, $P = 0.007$) compared with saline+DMSO (0.42 ± 0.05 mm³; Kruskal–Wallis test, $P = 0.01$) and saline+gefitinib (0.36 ± 0.04 mm³) groups, but it returned to normal values in the KA+gefitinib group (0.46 ± 0.02 mm³; Kruskal–Wallis test, $P = 0.001$; Fig S3A and B). Finally, we analyzed the effect of gefitinib in MTLE mice in terms of cell proliferation by administering BrdU three times (3 h apart) at 3 d post-KA injection (3dpKA) (Fig 3C). We found that the number of BrdU+ React-NSCs in the MTLE mice treated with gefitinib was significantly reduced compared with the DMSO-treated MTLE mice (a drop to 47.55 ± 9.62%; $t$ test, $P = 0.005$;

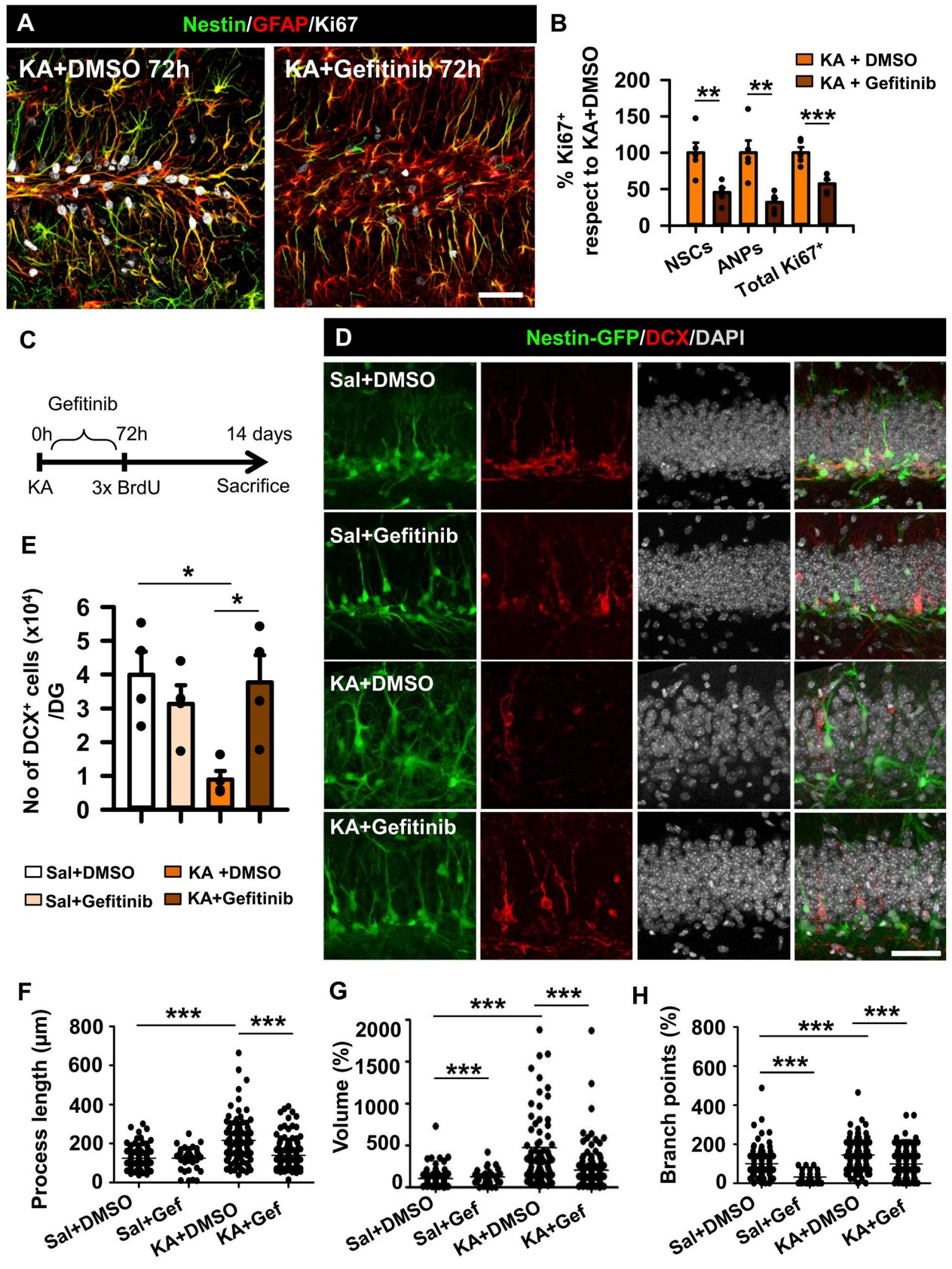

Fig S3D). Although diminished BrdU+ ANPs and total BrdU cells were observed, they did not reach significance, suggesting a more specific effect of EGFR blocking in NSCs in terms of proliferation.

### Heparin-binding epidermal growth factor (HB-EGF), a natural ligand of EGFR, increases early after MTLE-HS in the hippocampus

We next sought to investigate the potential natural contributors to the seizure-induced activation of EGFR in the neurogenic niche. We started exploring the expression of HB-EGF, a natural ligand of EGFR that has been recently reported to be a strong mitogenic factor for astrocytes (Jia et al, 2018) and is also able to stimulate astroglial migration (Faber-Elman et al, 1996). Furthermore, HB-EGF expression has been shown to be up-regulated in the hippocampus early after KA intraperitoneal administration (Opanashuk et al, 1999). We detected increased immunolabeling for HB-EGF 3dpKA in the SGZ and the portion of the molecular layer closer to the GCL of Nestin-GFP mice (Fig S3E). As immunohistochemical detection is not optimal for HB-EGF, we determined its levels by ELISA at different time points (1.5, 12, 24, and 72 h) after KA. HB-EGF showed a significant increase in the hippocampus 12hpKA and 24hpKA, decreasing to control levels at 72hpKA (876.35 ± 33.88 and 958.84 ± 26.45 pg/ml of HB-EGF at 12 and 24 h compared with 765.89 ± 20.06 pg/ml in animals injected with saline; one-way ANOVA, $P = 0.044$ and $P$ 0.001; Fig S3F). In addition, cultured NSPCs costained for HB-EGF (Fig S3G) and were able to release HB-EGF to the culture media (2,152.90 ± 202.07 pg/ml compared with 364.40 ± 2.77 pg/ml detected in control media [without cells] and 611.40 ± 1.39 pg/ml in the cellular pellet; Kruskal–Wallis test, $P = 0.02$ and $P = 0.05$; Fig S3H).

### Zinc, an activator of EGFR, increases early after seizures and regulates NSPC activity in vitro

We next sought to investigate the potential contribution of zinc release to the activation of the EGFR pathway. Zinc triggers EGFR signaling by binding not only directly to the receptor (Samet et al, 2003) but also indirectly by increasing HB-EGF release through the activation of zinc-dependent metalloproteinases (Wu et al, 2004). In addition, the over-release of zinc is a typical effect of neuronal hyperexcitation (Mody & Miller, 1985; Kasarskis et al, 1987). Reactive (free) zinc levels can be measured histochemically in control and MTLE-HS Nestin-GFP mice by the Danscher staining (Fig 4A). Saline-injected animals showed the characteristic distribution of zinc restricted to the hilar region with an average of 107.90 ± 9.19 zinc granules per Nestin-GFP+ cell, whereas KA-injected animals showed a significant increase of 188.00 ± 17.88 zinc granules per

Nestin-GFP+ cell ($t$ test, $P$ 0.001; Fig 4B). No difference was found in the size of the granules, a measurement used as an internal control for the technique (Fig 4B). To corroborate these findings, we immunostained against zinc transporter (ZnT3), which introduces zinc ions into synaptic vesicles (Wenzel et al, 1997; Cole et al, 1999) after 3dPKA. We observed a marked increase in the ZnT3 staining signal intensity (by pixel) in the granule cell layer from 14,997.50 ± 4,041.08 for saline- to 60,072.00 ± 7,985.31 for KA-injected mice. Furthermore, ZnT3 signal intensity overlapping with Nestin+GFP increased from 9,462.90 ± 3,857.63 to 22,628.57 ± 7,620.12. In the hilus, signal intensity changed from 76,090.87 ± 20,042.37 to 195,283.71 ± 28,167.48 for saline- and KA-injected mice, respectively ($t$ test, $P$ 0.001, $P$ 0.05, and $P$ 0.01; Fig S4A and B), in agreement with the Danscher staining data.

Furthermore, because zinc homeostasis in the CNS is controlled by metallothioneins (MTH), a family of metalloproteins responsible for buffering the level of intracellular labile zinc (Kagi & Schaffer, 1988), we tested their hippocampal expression in MTLE-HS. qRT–PCR analyses determined an increase in metallothionein I (MTH1) and a tendency to increase in II (MTH2) 24–72 h after KA administration (8.34 ± 2.07– and 10.89 ± 6.14–fold for MTH1 and 15.25 ± 4.20– and 19.42 ± 9.23–fold for MTH2; Kruskal–Wallis test, $P = 0.045$ for MTH1 and one-way ANOVA, $P = 0.087$ for MTH2; Fig S4C). Moreover, the presence of the proteins was confirmed by WB (Fig S4D). No changes were detected regarding the expression of MTH3. These data further support the increased presence and potential early involvement of zinc in the neurogenic niche in MTLE-HS.

### Zinc mimics the early effect of KA on the hippocampal neurogenic niche

We then moved to an in vivo experiment to further corroborate the effects of zinc on the neurogenic niche. We injected intra-hippocampally a single dose of 5, 20, or 30 nmol of zinc, with the first two doses reported to be tonic levels of zinc (Frederickson et al, 2006), and checked the effects after 1 wk. 24 h before euthanasia, we injected BrdU intraperitoneally to label proliferating cells (Fig 4C). We did not find an effect at 5 nmol (data not shown), but GCD and more BrdU+ cells, as well as increased Nestin-GFP expression, were observed in the DG using the 20 or 30 nmol doses recapitulating some of the effects caused by the intrahippocampal injection of 1 nmol of KA (MTLE-HS model) (Fig 4D). We focused on the 20-nmol dose and determined first an increase in the density of BrdU+ cells that incorporated in the SGZ+ GCL, rising from 26,038 ± 6,395 cells per millimeter$^3$ (saline) to 78,731 ± 9,036 cells per millimeter$^3$ (20 nmol zinc), a value close to 107,775 ± 6,245, found in the 1 nmol KA group (one-way ANOVA, $P$ 0.001, KA versus saline; $P = 0.001$, 20 nmol

**Figure 3. Inhibition of EGFR signaling with gefitinib in vivo reduces the induction of React-NSC induction and partially restores neurogenesis in MTLE-HS.**
**(A)** Confocal microscopy images of MTLE-HS mice treated with intranasal gefitinib (10 mg/Kg) or vehicle, 3 d after intrahippocampal KA injection (1 nmol) after staining for nestin, GFAP, and Ki67. Scale bar, 50 $\mu$m. **(B)** Quantification of proliferating (Ki67+) NSCs, ANPs, and overall proliferation in the SGZ expressed as the percentage with respect to the control. NSCs were identified as cells being immunopositive for nestin and GFAP and with a radial apical process extending across the GCL from the SGZ to the molecular layer. ANPs were located in the SGZ, were nestin-positive but GFAP-negative, and bore few and short processes. **(C)** Experimental design to treat the animals. **(D)** Confocal microscopy images of DG sections after immunostaining for GFP; DAPI was used to label all cell nuclei. Scale bar, 20 $\mu$m. **(F)** Quantification of changes in the length of the processes. **(G)** Quantification of changes in the cell volume. **(H)** Quantification of changes in the number of branch points. **(E)** Quantification of the volume of the GCL in saline+DMSO–, saline+gefitinib–, KA+DMSO–, and KA+gefitinib–treated animals 14dpKA. Data information: for (B), $t$ test, **$P$ < 0.01. Bars show the mean ± SEM. Dots show individual data. n = 4. For **(F, G, H)**, changes were quantified after 3D Sholl analysis and expressed as the percentage with respect to the control condition (saline+DMSO), one-way ANOVA, ***$P$ < 0.001. Dots show individual cells (F, G, H). For (E), Kruskal–Wallis test, *$P$ < 0.05, **$P$ < 0.01, and ***$P$ < 0.001. Bars show the mean ± SEM. Dots show individual data. n = 3.
Source data are available for this figure.

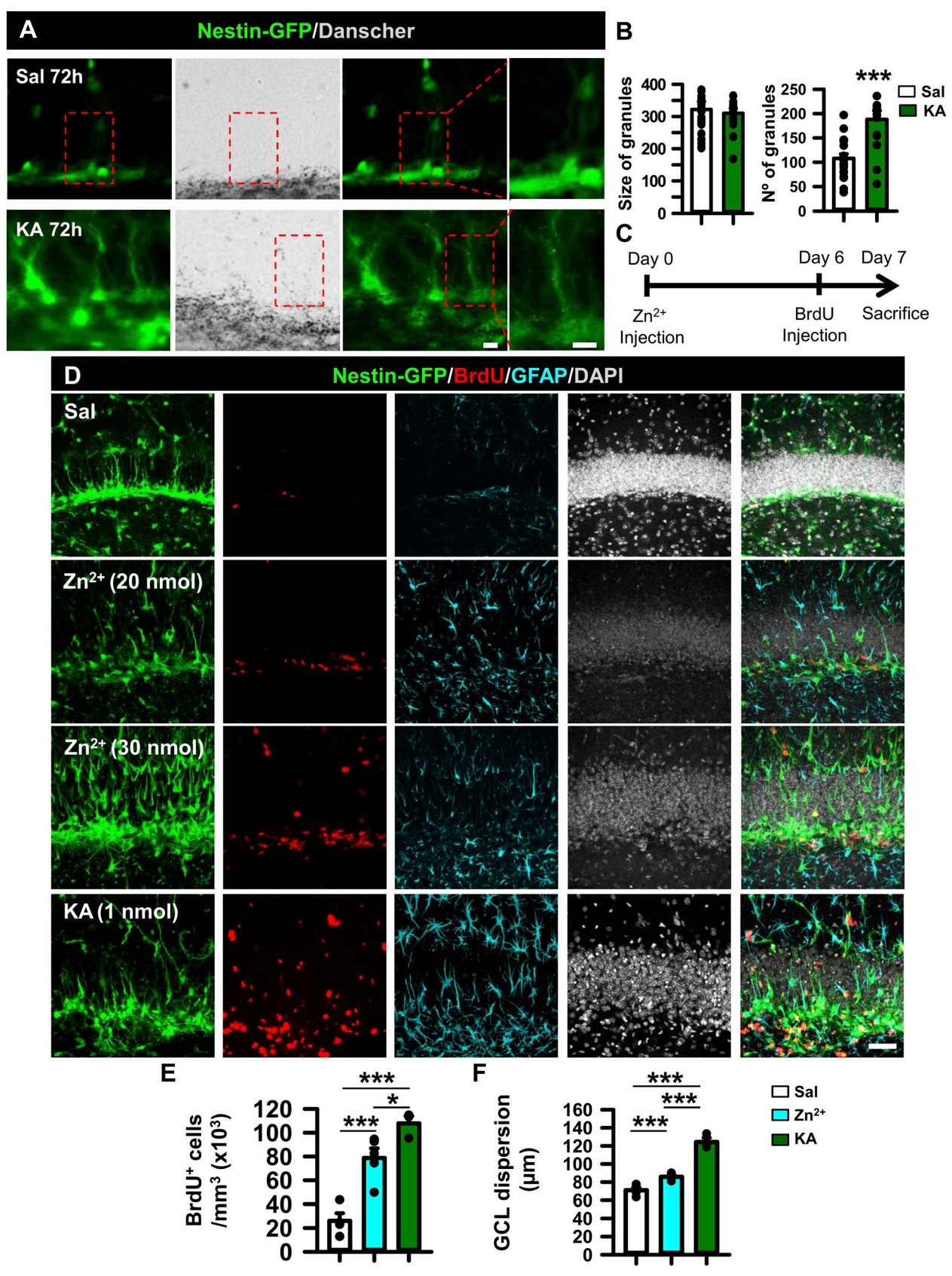

Zn+2 versus saline, and $P$ = 0.027, 20 nmol Zn+2 versus KA; Fig 4E). A clear induction of React-NSCs was observed, and a significant induction of GCD was measured: 70.96 ± 3.14 $\mu$m (saline), 85.75 ± 1.78 $\mu$m (20 nmol Zn+2), and 124.210 ± 4.67 $\mu$m (1 nmol KA) (one-way ANOVA, $P$ 0.001; Fig 4F). These results show that an intrahippocampal zinc injection mimics the disruption of the neurogenic niche triggered by KA administration. As mentioned before, an excess of zinc might have a positive effect by reducing neuronal hyperexcitation (Minami et al, 2006), and therefore, in our context of interest it could be argued that zinc chelation could be used to preserve the neurogenic niche in MTLE-HS. To test this hypothesis, we administered subcutaneously 5 mg/Kg of the zinc-chelating agent TPEN as previously described (Kim et al, 2012) on both control and MTLE-HS Nestin-GFP mice twice a day during seven consecutive days starting the same day that saline or KA was injected into the hippocampus to induce MTLE-HS. Although we did not find evident variations in GCD irrespective of the treatment (Fig S4E), we quantified the number of dying cells by assessing condensed DAPI staining, expecting a significant increase after KA administration as previously reported (Sierra et al, 2015) and evaluating the effect of zinc chelation by TPEN (Fig S4F). We observed a tendency in GCD and in cell death to increase after KA compared with saline-injected animals in both vehicle- and TPEN-treated mice. However, TPEN failed to alter cell death, as a similar density of pyknotic nuclei was found in the GCL than KA+vehicle–treated animals and significantly higher than saline+TPEN–treated animals (24,864.24 ± 630.84 pyknotic nuclei per millimeter[3] for KA+TPEN with respect to 2,895.61 ± 1,635.08 pyknotic nuclei per millimeter[3] for saline+TPEN; Kruskal–Wallis test, $P$ = 0.014; Fig S4F). These results suggest that despite mimicking the early effects of MTLE-HS on the neurogenic niche, endogenous zinc does not contribute to cell death.

### HB-EGF and zinc activate the EGFR pathway in NSPCs in vitro

To confirm whether zinc could modulate NSPCs in an EGFR-mediated manner, we cultured NSPCs from the hippocampus of Nestin-GFP mice. To avoid EGFR activation caused by growth factors present in the culture media, NSPCs were subjected to growth factor starvation for 15 min, 2 h, or 6 h. A marked loss of phosphorylated ERK (p-ERK) was found at the 2-h time point (Fig S5A). Following a protocol adopted from Wu et al (2004), NSPCs were exposed to 100 ng/ml EGF or 200 $\mu$M zinc, or pretreated with 2 $\mu$M gefitinib for 60 min before the addition of 200 $\mu$M zinc. Protein extracts were blotted sequentially against Tyr845 and Tyr1068 phosphorylation sites of EGFR and total EGFR to test receptor phosphorylation (Figs 5A and S5B). Although we could not fully exclude that autocrine-released HB-EGF could trigger some residual EGFR phosphorylation, the starving condition was

considered the control as with the lowest phosphorylation (ratio to control = 1). We observed that zinc induced a mild phosphorylation of Tyr845 that was fully blocked in the presence of gefitinib (ratio 1.00 ± 0.13 [starved] compared with 2.36 ± 0.20 [zinc]; Kruskal–Wallis test, $P$ = 0.05; and 2.36 ± 0.20 [zinc] compared with 0.42 ± 0.04 [zinc+gefitinib]; Kruskal–Wallis test, $P$ = 0.02; Fig 5B and C). However, there was no statistical difference for Tyr1068 phosphorylation (Fig S5C). It has also been reported that a feedback effect by which p-ERK mediates HB-EGF shedding and subsequent EGFR activation can take place (Yin & Yu, 2009). We observed that zinc exposure in starved conditions was able to increase p-ERK signaling, and interestingly, the inhibition of EGFR reduced its phosphorylation (ratio 1.00 ± 0.04 [starved] compared with 2.23 ± 0.16 [zinc]; one-way ANOVA, $P$ = 0.017; and 2.23 ± 0.16 [zinc] compared with 0.85 ± 0.35 [zinc+gefitinib]; one-way ANOVA, $P$ = 0.015; Fig 5B and D). To further corroborate this effect and to get closer to in vivo conditions, we stimulated starved cells with the highly mitotic HB-EGF ligand, HB-EGF+zinc, or pretreated cells with gefitinib and then exposed them to HB-EGF+zinc. Our results demonstrated that HB-EGF strongly phosphorylated EGFR at both Tyr845 and Tyr1068 residues. Remarkably, zinc stimulation did not produce any significant additive effect, possibly because of the excess of HB-EGF. Importantly, gefitinib pretreatment blocked phosphorylation at Tyr845 and Tyr1068 (ratio 1.00 ± 0.29 [starved] compared with 3.12 ± 0.52 [zinc+HB-EGF]; one-way ANOVA, $P$ = 0.027; and 3.12 ± 0.52 [zinc+HB-EGF] compared with 1.09 ± 0.39 [zinc+HB-EGF+gefitinib]; one-way ANOVA, $P$ = 0.046 for Tyr845; and ratio 1.00 ± 0.16 [starved] compared with 4.63 ± 0.89 [zinc+HB-EGF]; one-way ANOVA, $P$ = 0.005; and 4.63 ± 0.89 [zinc+HB-EGF] compared with 0.51 ± 0.14 [zinc+HB-EGF+gefitinib]; one-way ANOVA, $P$ = 0.002 for Tyr1068; Fig 5E–G). Moreover, EGFR inhibition showed a tendency to reduce downstream p-ERK even in the concomitant presence of both zinc and HB-EGF (Fig 5E and H). These results suggested that inhibiting EGFR with gefitinib was sufficient to inactivate the receptor regardless of the autocrine and paracrine HB-EGF–dependent phosphorylation.

### Zinc induces React-NSCs through activation of EGFR in vivo

The next step was to examine the in vivo effect of EGFR inhibition in animals that received a single dose of zinc (20 nmol) intrahippocampally, expecting that blocking EGFR would reduce the zinc-induced overproliferation of cells. For this purpose, we administered intranasal gefitinib during the first 72 h following the same paradigm described previously. The animals received three doses (3 h apart) of BrdU intraperitoneally to track dividing cells starting 24 h before euthanasia (Fig 5I). Our results showed that EGFR inhibition reduced the increase in BrdU+ cells in the GCL provoked by intrahippocampal

---

**Figure 4. Zinc mimics the effect of seizures in the hippocampal neurogenic niche.**
**(A)** Danscher's staining 3 d after saline or KA intrahippocampal injection in Nestin-GFP mice showing the wider presence of zinc in the SGZ and GCL. Scale bar, 10 $\mu$m. **(B)** Quantification of the number of precipitated zinc granules. The size of the granules was assessed as an internal control. **(C)** Schematic of the in vivo zinc intrahippocampal injection and BrdU administration. Nestin-GFP mice were euthanized 7 d after zinc or KA 1 nmol administration (20 nmol), and BrdU was given 24 h before euthanasia to identify mitotic cells. **(D)** Confocal microscopy images after immunostaining for GFP, BrdU, and GFAP. DAPI was used as nuclear staining. Increased cell proliferation, GCD, and induction of React-NSCs are noticeable. Scale bar, 10 $\mu$m. **(E)** Quantification of the density of BrdU+ cells in the SGZ+GCL. **(F)** Quantification of GCD. Data information: for (B), $t$ test, ***$P$ < 0.001. Bars show the mean ± SEM. Dots show individual data. For (E, F), one-way ANOVA, *$P$ < 0.05 and ***$P$ < 0.001. Dots show individual cells (E, F). n = 3–4. Source data are available for this figure.

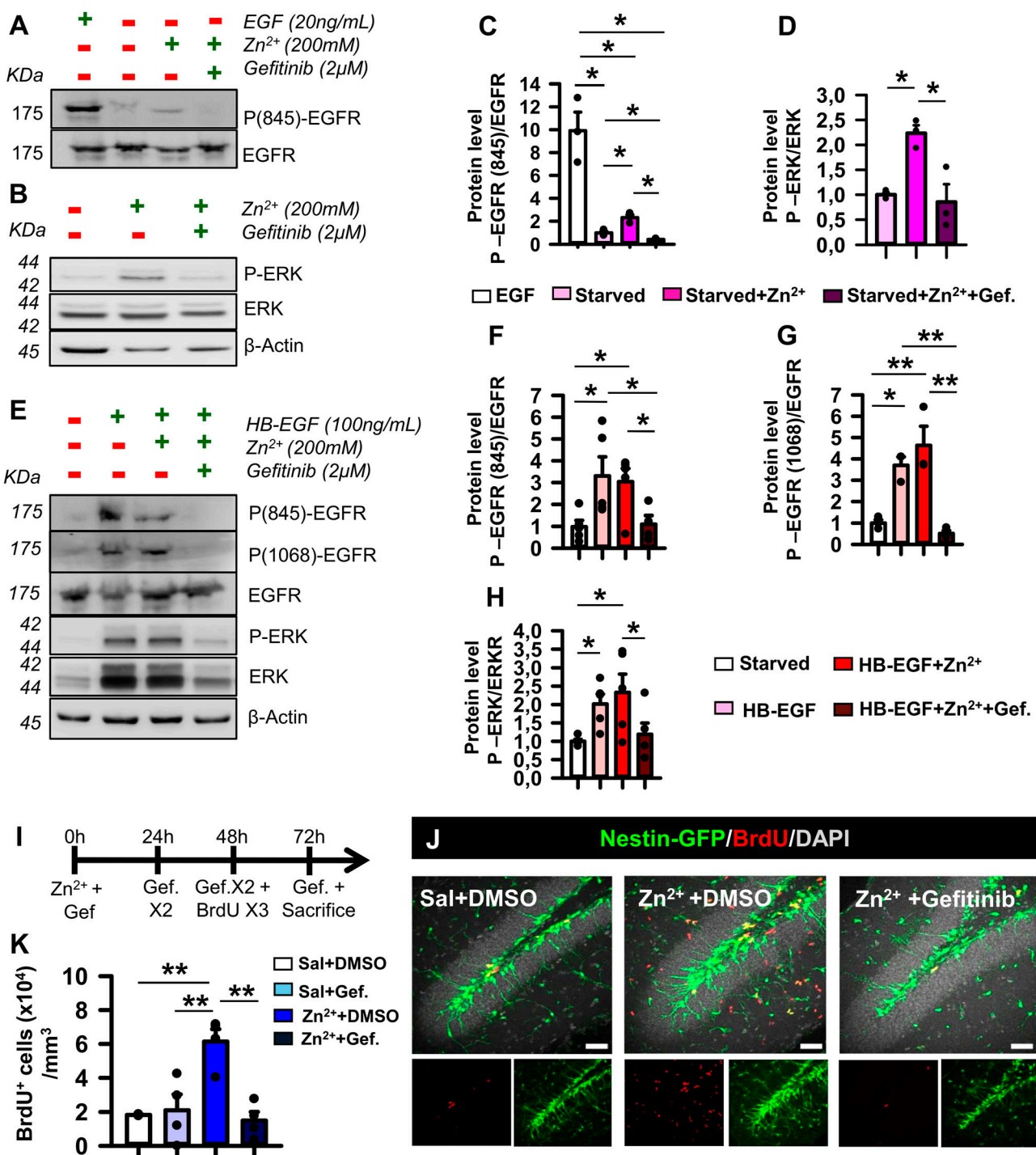

**Figure 5. Gefitinib blocks the zinc-induced activation of the EGFR signaling pathway in NSPCs.**
**(A)** Pretreatment with gefitinib (2 μM) prevents the phosphorylation of EGFR induced by EGF and by zinc (200 μM) as shown by WB. **(B)** WB of in vitro–cultured NSPCs showing that activation of p-ERK downstream signaling in the presence of zinc (200 μM) is blocked by gefitinib (2 μM). Note that β-actin shows no variation in loading inputs. **(C)** Ratio of phosphorylated to non-phosphorylated EGFR Tyr845. **(D)** Ratio of phosphorylated to non-phosphorylated ERK. **(E)** Presence of HB-EGF and zinc stimulates EGFR phosphorylation at Tyr845 and Tyr1068 sites and shows a tendency to stimulate p-ERK signaling. Gefitinib reduces both phosphorylation of EGFR and p-ERK. **(F)** Quantification of the phosphorylated to the non-phosphorylated ratio for EGFR Tyr845. **(G)** Quantifications of the phosphorylated to the non-phosphorylated ratio for EGFR Tyr1068. **(H)** Quantifications of the phosphorylated to the non-phosphorylated ratio for ERK. **(I)** Schematic of intranasal gefitinib treatment after intrahippocampal zinc injection. BrdU was given 24 h before euthanasia to identify mitotic cells. **(J)** Confocal microscopy images of saline, zinc, and zinc+gefitinib (10 mg/Kg) 3 d after zinc showing the increase in BrdU+ cells in the SGZ of zinc-injected animals, which is reversed by the administration of gefitinib. Scale bar, 50 μm. **(K)** Quantification of total BrdU+ cells in the SGZ. There is a significant increase in BrdU+ cells in the SGZ after zinc administration, which, however, contained administering gefitinib. Data information: for (C), Kruskal–Wallis analysis, *P < 0.05. For (D), one-way ANOVA, *P < 0.05. Bars show the mean ± SEM. Dots show individual data (C, D). For (F, G, H),

injection of zinc (15,680 ± 2,590 [saline] compared with 51,360 ± 7,890 [zinc]; one-way ANOVA, $P$ = 0.016; and 51,360 ± 7,890 [zinc] compared with 23,970 ± 6,830 [zinc+gefitinib]; one-way ANOVA, $P$ = 0.026; Fig 5J and K). No significant changes in the expression and distribution of EGFR by NSCs were found 24 h after zinc injection (Fig S5D and E).

To check the effect on early neurogenesis, we repeated the experiment but euthanized the animals at 14 d post-intrahippocampal zinc injection. Despite the increase in proliferation and the drastic change of the neurogenic niche in the short term, similar to that of MTLE, we observed no ablation of DCX+ cells (Fig S5D and E). On the contrary, the results showed a tendency for neurogenesis to increase, expressed as the ratio of BrdU+DCX+ cells to DCX+ cells (Fig S5F). These results support the notion that zinc is another contributor to the complex responses triggered by seizures rather than a single master effector. As an example, no significant changes in the expression and distribution of EGFR by NSCs were found 24 h after zinc injection (Fig S5G and H).

# Discussion

### Seizures induce the transformation of hippocampal NSCs into gliogenic React-NSCs

A population of radial glia-like NSCs was shown to be the source of postnatal and adult neurogenesis in the hippocampus in pioneering work that also suggested their capability to generate astrocytes (Seri et al, 2001). This astrogliogenic capability was later confirmed (Encinas et al, 2011), along with their capacity to generate additional NSCs (Bonaguidi et al, 2011; Pilz et al, 2018), establishing the natural multipotent capability of NSCs in physiological conditions. More recently, the cell differentiation diversity of hippocampal NSCs was further expanded, as we showed that seizures trigger the conversion of NSCs into reactive astrocytes through an intermediate cell type, termed React-NSCs, which is distinct from both NSCs and reactive astrocytes (Sierra et al, 2015; Muro-García et al, 2019). Similarly, it was reported that NSCs from the SVZ generate reactive astrocytes that migrate into the cortex after experimental stroke (Faiz et al, 2015). Complete ablation of neurogenesis might be a specific outcome of MTLE with HS, whereas in other models, a boost in neurogenesis with abnormal features (aberrant neurogenesis) is generally observed (Kuruba et al, 2009; Pineda & Encinas, 2016). A direct comparison among different models (intracerebral KA, intraperitoneal KA, pilocarpine, electrical kindling, etc.) is challenging because of variations in paradigms, points of analysis (cell markers, proliferation, survival, differentiation), and the different cell types studied (NSCs, newborn neurons, astrocytes, etc.). The mechanisms driving increased aberrant neurogenesis in MTLE versus the absence of neurogenesis in MTLE-HS deserve further investigation. Currently, we can speculate that different levels of local neuronal hyperexcitation and the presence or absence of neuronal death and reactive gliosis in the hippocampus may trigger qualitatively different

responses in the neurogenic niche, reflecting the high level of plasticity of NSCs and neurogenesis (Bielefeld et al, 2019).

Characterizing the active role of NSCs in the brain's response to damage by contributing to reactive gliosis opens new avenues for potential therapeutic interventions. Unveiling the mechanisms controlling the transformation of NSCs to React-NSCs might allow us not only to preserve healthy NSCs and eventually neurogenesis in MTLE-HS but also to manipulate the level of reactive gliosis. Herein, we demonstrate that EGFR is a key regulator in the induction of React-NSCs after seizures and that its blockade can preserve neurogenesis. Furthermore, we show that not only HB-EGF, the natural ligand of EGFR, but also zinc is involved in the EGFR-dependent induction of React-NSCs.

### Intrahippocampal KA injection triggers the overexpression of EGFR in hippocampal NSCs. Blocking EGFR with gefitinib partially preserves NSCs

In vitro studies using stem cell–like germinative cells highlighted the EGFR/ERK transduction pathway as responsible for activating and inducing proliferation upon the addition of human EGF, with EGFR inhibition blocking cell proliferation (Cheng et al, 2017). Other studies have shown that inactivation of the EGFR/MAPK pathway blocks or greatly retards NSC cycle progression in Drosophila (Li et al, 2015), and the absence of EGFR strongly impairs stem cell self-renewal (Robson et al, 2018). EGFR has been shown to be expressed in activated and proliferating NSPCs (Okano et al, 1996; Jhaveri et al, 2015), and is used to distinguish activated NSPCs from quiescent ones (Walker et al, 2016). In addition, EGFR activation controls the transformation of astrocytes into hypertrophic reactive astrocytes (Liu & Neufeld, 2004; Liu et al, 2006; Tsugane et al, 2007), acting as an upstream signal for exiting quiescence (Liu et al, 2006). Therefore, we hypothesized that EGFR could regulate the transformation of NSCs into React-NSCs, a transient cell type with features of both NSCs and reactive astrocytes. We confirmed that soon after the induction of MTLE-HS, EGFR was overexpressed in the hippocampus, particularly in the neurogenic niche, where it colocalized with Nestin-GFP+ NSCs and ANPs. Downstream effectors of EGFR, such as phospho-STAT3 and phospho-ERK1/2, were also elevated. Blocking EGFR with gefitinib in NSPCs derived from the hippocampus reduced cell proliferation, a result that was replicated in vivo. Importantly, gefitinib not only reduced the seizure-induced increase in NSC proliferation, but also, at least partially, preserved the healthy morphology of NSCs.

### HB-EGF, a natural ligand of EGFR, is also overexpressed after seizures, and its interplay with over-released zinc contributes to EGFR activation after seizures

We explored the role of HB-EGF, a natural ligand of EGFR. Binding of HB-EGF to EGFR has been associated with enhanced EGFR tyrosine kinase activity and prolonged ERK activation (Yoo et al, 2012). Our

---

Kruskal–Wallis analysis, *$P$ < 0.05 and **$P$ < 0.01. Bars show the mean ± SEM. Dots show individual data (F, G, H). For (K), one-way ANOVA, *$P$ < 0.05. Bars show the mean ± SEM. Dots show individual data. n = 3.
Source data are available for this figure.

results also show that there is an early increase in the expression of HB-EGF, the natural ligand of EGFR, which occurs in parallel with the increase in EGFR expression. This finding is consistent with the previous work based on intraperitoneal administration of KA, where HB-EGF mRNA increased within 3 h and remained elevated for at least 48 h in the DG (Opanashuk et al, 1999). EGFR and HB-EGF have been associated with pro-inflammatory cytokine release (Richter et al, 2002), and HB-EGF induced astrocyte proliferation in vitro in an EGFR-dependent manner. Therefore, HB-EGF could also be a potential therapeutic target. However, because it also enhances neuronal and astrocyte survival (Kornblum et al, 1999; Jia et al, 2018) and is neuroprotective against excitotoxicity (Opanashuk et al, 1999), this option is not without concern.

We were intrigued by the potential role of zinc in the activation of the EGFR pathway for several reasons: (1) HB-EGF is initially synthesized as a membrane-bound precursor (pro-HB-EGF), and it is cleaved at the juxtamembrane domain to release the soluble form of HB-EGF by matrix metalloproteinases (Izumi et al, 1998), which are activated by zinc (Le Gall et al, 2003; Wu et al, 2004); (2) zinc-dependent ERK activation can stimulate EGFR phosphorylation through HB-EGF (Yin & Yu, 2009); and (3) zinc is massively released from excitatory terminals during neuronal hyperactivity, reducing neuronal activation (Assaf & Chung, 1984). In the brain, the highest concentration of zinc has been found in the mossy fibers (MFs) of granule cells (Frederickson, 1989). MFs connect to CA3 pyramidal neurons and dentate hilar cells. However, after the KA lesion, the MFs sprout in the DG, forming new synapses on granule cell dendrites, which increases the excitatory connections between granule cells (Buckmaster et al, 2002) and seizure susceptibility (Sutula et al, 1989; Shetty et al, 2005). Knockout mice lacking the synaptic zinc transporter have increased excitotoxicity after KA-induced neuronal hyperexcitation (Cole et al, 2000). In addition, zinc chelation was able to provoke the induction of seizures, producing paroxysmal epileptiform brain activity and subsequent damage (Mitchell & Barnes, 1993; Cuajungco & Lees, 1998). Furthermore, chelation of zinc during neural hyperexcitation exacerbated excitotoxicity (Domínguez et al, 2003, 2006). Other studies suggested that zinc released into the extracellular space by glutamatergic synapses could be toxic (Babb et al, 1991; Lee et al, 2000). Our data reported herein support the detrimental effect of zinc chelation, as it exacerbated cell death in MTLE-HS. Nevertheless, we found that intrahippocampal zinc administration recapitulated several effects of KA injection as a model of MTLE-HS. Intra-hippocampal zinc injection triggered massive proliferation and reactive gliosis, including the induction of React-NSCs in the neurogenic niche and GCD in a dose-dependent manner. Interestingly, the extracellular matrix plays an important role in regulating physiological plasticity during epileptogenesis (Dityatev & Fellin, 2008). Extracellular matrix disintegrins and metalloproteases contain a zinc-binding motif critical for proteinase activity (Sagane et al, 1998). It is plausible to speculate that an alteration in their function could affect the proteolytic cleavage of proteins such as Reelin, which has a direct role in neuronal migration affecting GCD (Tinnes et al, 2011). In cultured NSPCs, zinc (and HB-EGF) increased cell proliferation and activated the EGFR and its transduction pathway. To test the hypothesis that zinc could be directly acting on EGFR, we again used gefitinib in vitro and in vivo. Gefitinib indeed

blocked the effects of zinc in cultured NSPCs in vitro and in the neurogenic niche in vivo, leading us to conclude that the excess of zinc released during seizures contributes to the induction of React-NSCs by directly activating EGFR, which NSCs overexpress because of neuronal hyperexcitation.

Regulating zinc concentration is therefore a key event that can dictate the final effect of zinc in neuronal hyperexcitation. Intracellular zinc buffering is largely regulated by a family of small cysteine-rich proteins called metallothioneins (MTH) (Ebadi & Hama, 1986). We found that MTH1 mRNA was significantly increased at 24 h post-KA administration, and there was a tendency for MTH2 to increase from 12 h onward, highlighting that the triggering of environmental changes mediated by zinc release occurs very early and probably plays a role in the shedding and release of diffusible HB-EGF.

### Blocking EGFR directly, rather than targeting HB-EGF or zinc, is a better strategy for preserving NSCs

Overall, our results strongly suggest that directly targeting EGFR, rather than the release of zinc or HB-EGF, is a more effective strategy for potentially preserving NSCs and neurogenesis. Treatment with EGFR inhibitors has been reported as neuroprotective in rat models of glaucoma (Liu et al, 2006) and spinal cord injury (Erschbamer et al, 2007) by acting preferentially on reactive astrocytes. Our results show that EGFR inhibition was able to block ERK1/2 phosphorylation in NSPCs, suggesting the involvement of EGFR in the mitogenic ERK1/2 pathway in NSCs. Although gefitinib strongly impaired cell division, as assessed by BrdU incorporation in treated NSPC cultures, we still observed a fraction of cells undergoing active nucleotide incorporation, and we cannot exclude the concurrent involvement of other signaling pathways. Irradiation-induced ablation of cell division enriched the fraction of EGFR-negative quiescent NSCs in the SVZ. However, NSCs were able to re-enter the cell cycle shortly after 24–48 h to repopulate the niche (Daynac et al, 2013), suggesting that other signaling pathways, such as Sonic hedgehog (Shh) (Daynac et al, 2016), also play an important role in NSC activation. In conclusion, our work demonstrates for the first time a molecular mechanism involved in the induction of React-NSCs and the subsequent loss of neurogenesis: the EGFR pathway. After seizures, EGFR expression increases specifically in NSCs in the hippocampal neurogenic niche, as does that of HB-EGF and the release of zinc. HB-EGF is a natural ligand of EGFR, and zinc can bind directly to EGFR to activate it. Moreover, zinc can potentiate the release of HB-EGF, further contributing to EGFR activation. By blocking EGFR with gefitinib, we were able to prevent the induction of React-NSCs, at least in terms of activation and morphology.

## Materials and Methods

### Mice

Experimental procedures were performed in compliance with the European Community Council Directive of 24 November 1986 (86/609/EEC) and were approved by the University of the Basque

Country (UPV/EHU) Ethics Committees and the Diputación Foral de Bizkaia. The protocol reference is CEEA M20/2015/236. The animals were maintained with access to food and water ad libitum in a colony room that was maintained at a constant temperature (19–22°C) and humidity (40–50%) on a 12:12-h light/dark cycle. Nestin-GFP (Mignone et al, 2004) and WT Nestin-GFP–negative littermates on C57BL/6J background counterparts were used. The animals were 2 mo old and weighed 25–30 g at the time of experimentation. Both males and females were indistinctly used in each experiment, and littermates were randomly assigned to each experimental condition.

## Cell cultures

Adult hippocampal NSPC cultures were obtained using an adaptation of protocols as previously described (Pineda et al, 2013; Jhaveri et al, 2015). NSPCs were isolated from either Nestin-GFP or WT littermates. Briefly, hippocampi were dissected from five adult mice and placed in ice-cooled PBS with sucrose (PBS: phosphate-buffered saline; Cat#806544 and sucrose; Cat#S0389; Sigma-Aldrich). The tissue was cut into chunks of 2–3 mm size that were pooled together and incubated with a mixture of papain (1 mg/ml papain (15 UI/ml), Cat#LS003126; Worthington) and DNase (2.5 $\mu$l DNase 2,000 U/ml, Cat#10636153; Thermo Fisher Scientific) for 30 min for enzymatic digestion. Enzymes were inactivated with ovomucoid solution (0.7 mg/ml, Cat#T9253; Sigma-Aldrich), and tissue was mechanically homogenized with a P1000 pipette tip. The resultant cell mixture was passed through a 20-$\mu$m Nylon filter (Falcon, Cat#Z290823; Agilent Technologies) and washed twice with BSA diluted with PBS at 0.15%. The cell mixture underwent centrifugation at 200 for 10 min and was then cultured using NeuroCult proliferation medium (Cat#05702; StemCell Technologies). The cells were maintained at standard conditions in a humidified 37°C incubator containing 5% $CO_2$. NSPC cultures were then passaged every 7 d by enzymatic disaggregation with Accutase (Cat#7,920; StemCell Technologies). The medium was supplemented with a NeuroCult proliferation supplement (Cat#05702; StemCell Technologies) at 9:1 ratio; heparin solution 2 $\mu$g/ml (Cat#07980; StemCell Technologies), EGF 20 ng/ml (Cat#315-09; Peprotech), and FGF2 10 ng/ml (Cat#450-33; Peprotech); and penicillin 100 U/ml and streptomycin 150 $\mu$g/ml (15140-122; Gibco). NSPCs were maintained for a maximum of four total passages to preserve a heterogeneous population and avoid cell selection issues. For proliferation/inhibition assays, 25,000 NSPCs per well were seeded into 12-mm coverslips coated with laminin (L2020; Sigma-Aldrich) as previously described (Silvestre et al, 2011), to avoid inhibition of cell proliferation by cell-to-cell contact. Cells were grown for 24 or 48 h, and a pulse of 10 $\mu$M BrdU (Cat#19-160; Sigma-Aldrich) for 1 h was given to detect cell proliferation. To assess either apoptotic or necrotic cell death, pyknotic nuclei were identified with 4',6-diamidino-2-phenylindole (DAPI) staining (Cat#D9542-5MG; Sigma-Aldrich).

## MTLE-HS model

For the stereotaxic injection of kainic acid (KA) (Cat#K0250; Sigma-Aldrich), animals were anesthetized with a mixture of ketamine (75 mg/Kg; Ketamine) and medetomidine (1 mg/Kg; Sedastart; Pfizer) A single

dose of the analgesic buprenorphine (1 mg/kg) (Buprecare, Animalcare Ltd) was administered subcutaneously. To induce status epilepticus, 50 nl of a 20 mM solution of KA was delivered at the following co-ordinates: anteroposterior (AP), −1.9; lateral (L), −1.5; dorsoventral (DV), −2 mm, with bregma as reference. The delivery of the solution was performed using a micropump (Nanoject II; Drummond Scientific) using a glass capillary as previously described (Sierra et al, 2015). Control mice were given an intrahippocampal injection of 50 nl of 0.9% sterile sodium chloride (NaCl) (Cat#S5886; Sigma-Aldrich). We confirmed the existence of initial seizures behaviorally using the Racine scale. All the animals included in the study reached category 4 or 5: clonic rearing and generalized tonic–clonic seizures with loss of righting reflex. All the MTLE mice reached at least one episode of category 4 or 5 convulsions during the period of 5–6 h in which they were monitored at the day of the surgery. The mortality rate is 10% with most of deaths occurring during or immediately after the surgery. The correct targeting of the injection was confirmed in the slices used for immunostaining.

## Treatments

The stereotaxic injection of intrahippocampal zinc was performed using the coordinates described above using a 33G stainless steel cannula (outer diameter, 0.28 mm) connected to a Hamilton microsyringe (Hamilton) as previously reported (Luzuriaga et al, 2019). The EGFR irreversible inhibitor afatinib dimaleate (Ref. S7810; Selleck Chemicals LLC) was diluted with saline to a concentration of 2 mg/Kg and injected intrahippocampally immediately before KA delivery following the same steps as for the KA injection. In all conditions, the glass capillary or the cannula was left in the hippocampus for an additional 5 min to avoid reflux. The skin of the skull was sutured, and the mice were maintained in a thermal blanket until they recovered from anesthesia. For zinc chelation, N,N,N,N-Tetrakis(2-pyridylmethyl)ethylenediamine (TPEN) (Cat#P4413-50MG; Sigma-Aldrich) solution was freshly prepared in 10% ethanol (Cat#100983; Merck) and saline, and injected subcutaneously at 5 mg/Kg, twice a day as previously described (Kim et al, 2012). For intranasal treatment, 10 mg/Kg of gefitinib (Ref. S1025-SEL; Selleck Chemicals LLC) was diluted with DMSO (Cat#D8418; Sigma-Aldrich) plus 0.5% carboxymethylcellulose (Cat#419273; Sigma-Aldrich) and 0.2% Tween-80 (Cat#P4780; Sigma-Aldrich) following the manufacturer's instructions. Gefitinib administration started immediately after the stereotaxic KA injection, and continued twice a day for the first 3 d as previously described (Hanson et al, 2013; Pineda et al, 2013). BrdU (Cat#19-160; Sigma-Aldrich) was diluted in sterile saline at 150 mg/kg concentration and administered through three intraperitoneal injections separated by 3-h intervals. BrdU injections were done 24 h before euthanasia for the analysis at 3dpKA and 3/7 d post-administration of zinc (Cat#83265-250ML-F; Sigma-Aldrich). To identify newborn cells at 14dpKA, BrdU was given on the last day of gefitinib treatment administration.

## Real-time quantitative polymerase chain reaction

Ipsilateral hippocampi from WT mice were dissected at different time points (1.5, 12, 24, and 72 h) post-KA injection, immediately

frozen with RLT buffer (Cat#79216; QIAGEN), and processed for total RNA extraction using a Micro RNeasy Plus isolation kit (Cat#74034; QIAGEN). Reverse transcription was done using a high-capacity reverse transcription kit (Cat#S7810; Thermo Fisher Scientific). A real-time quantitative polymerase chain reaction (qRT–PCR) was performed on a Bio-Rad CFX96 device (Bio-Rad), using the following primers:

Hypoxanthine–guanine phosphoribosyltransferase (HPRT) forward: 5′-GTT GGG CTT ACC TCA CTG CT-3′, reverse: 5′-TCA TCG CTA ATC ACG ACG CT-3′; glyceraldehyde phosphate dehydrogenase (GAPDH) forward: 5′-CCA GTA TGA CTC CAC TCA CG-3′, reverse: 5′-GAC TCC ACG ACA TAC TCA GC-3′; EGFR forward: 5′-GCC AAC TGT ACC TAT GGA TGT-3′, reverse: 5′-GGC CCA GAG GAT TTG GAA GAA-3′; FGFR 1 forward: 5′-CCA AAC CCT GTA GCT CCC TA-3′, reverse: 5′-TGA ACT TCA CCG TCT TGG CA-3′; FGFR2 forward: 5′-CCG AAT GAAGAC CAC GAC CA-3′; reverse: 5′-TCG GCC GAA ACT GTT ACC TG-3′; metallothionein (MTH) 1 forward: 5′-TCA CCA CGA CTT CAA CGT CC-3′, reverse: 5′-CAG TTG GGG TCC ATT CCG AG-3′; MTH2 forward: 5′-GCA TCT GCA AAG AGG CTT CC-3′, reverse: 5′-AGT TGT GGA GAA CGA GTC AGG-3′; MTH3 forward: 5′-GCT GCT GGA CTG GAT ATG GA-3′; reverse: 5′-TTG CAT TTG TCC GAG CAG GT-3′. Each sample was normalized to endogenous GAPDH and to HPRT that were used as housekeeper genes. Each reaction was performed at least twice in duplicates, and the relative expression of each gene was calculated using the standard $2^{-\Delta\Delta Ct}$ method (Livak & Schmittgen, 2001). HPRT and GAPDH were used as housekeeper genes.

## Tissue and cell fixation and processing

Nestin-GFP mice or their WT (Nestin-GFP–negative) control and treated animals were deeply anesthetized with an intraperitoneal overdose of 2.5% of 2,2,2-tribromoethanol (Avertin) (Cat#T48402-25G; Sigma-Aldrich) and were subjected to intracardiac perfusion with 30 ml PBS followed by 30 ml 4% PFA (Cat#158127; Sigma-Aldrich) in 0.1 M PBS (pH 7.4). The brains were dissected and post-fixed for an additional 3 h at room temperature in 4% PFA and then rinsed with PBS. Serial sagittal vibratome sections were made (50 $\mu$m thick) using a Leica VT 1200S vibrating blade microtome (Leica Microsystems GmbH) and stored with PBS/0.02% sodium azide (Cat#S2002; Sigma-Aldrich) at 4°C until use. The ipsilateral hemisphere was sliced sagittally in a lateral-to-medial direction, from the beginning of the lateral ventricle to the middle line, thus including the entire dentate gyrus (DG). The 50-$\mu$m slices were collected in six parallel sets, each set consisting of 12 slices, each slice 300 $\mu$m apart from the next. Cell cultures at the end of the treatments were fixed with PFA 4% dissolved with PBS/4% sucrose, rinsed with PBS, and stored with PBS/0.02% sodium azide until use.

## Immunofluorescence

For immunofluorescence, sections were incubated with blocking and permeabilization solution (PBS containing 0.25% Triton X-100 (Cat#93443; Sigma-Aldrich) and 3% BSA (Cat#A2153; Sigma-Aldrich)) for 3 h at room temperature and then incubated overnight with the primary antibodies (diluted in the same solution) at 4°C. Cell cultures were permeabilized with PBS/0.3% Triton X-100/1% BSA and then incubated with the respective primary antibodies

overnight at 4°C. For BrdU and nestin staining, pretreatment with 2M hydrochlorhydric acid (HCl; Cat#1003141000; Sigma-Aldrich) was applied during 20 min at 37°C followed by an immediate incubation with 0.1 M tetraborate (Cat#221732; Sigma-Aldrich) for 10 min at room temperature before the blocking and permeabilization step. After the incubation, the primary antibody was removed and the sections or cell-containing coverslips were washed with PBS three times for 10 min. Next, they were incubated with fluorochrome-conjugated secondary antibodies diluted in the permeabilization and blocking solution for 3 h at room temperature. After washing with PBS, the sections or cell-containing coverslips were mounted on gelatin-coated slides with Fluorescent Mounting Medium (S3023; Dako Cytomation). The following primary antibodies were used: GFP (1:1,000, #GFP-1,020) and nestin (1:1,000, #Nes) from Aves Laboratories; FGFR1 (1:200, #9740) and HB-EGF (1:400; AF8239-SP) from Cell Signaling Technologies; EGFR (1:1,000, ab52894) from Abcam; Ki67 (1:400) from Vector Laboratories; NeuN (1:200), Ki67 (1:1,000), and GFAP (1:1,500) from Dako Cytomation; BrdU (1:300, Ab6326) from AbD Serotec; and ZnT3 (1:500, 17363-1-AP) from Thermo Fisher Scientific–Proteintech, and the following secondary antibodies were used: donkey anti-mouse, anti-rabbit, anti-goat, or anti-rat Alexa 488, 568, 594, 647, or 680 secondary antibodies from Thermo Fisher Scientific; anti-chicken secondary IgG fluorescein (FITC; 603-702-C37; Tebu-bio) DAPI at 1:1,000 (Cat#D9542-5MG; Sigma-Aldrich) was used at counterstaining (for cell nuclei) when required.

## 3D Sholl analysis

3D images containing single cells (at least n = 50 per condition) were generated from z-stacks of saline+saline (n = 4), saline+gefitinib (n = 3), KA+saline (n = 3), and KA+gefitinib (n = 5) Nestin-GFP mice using a confocal microscopy. The cell profile was delimited using the "polygon" tool of ImageJ, and a mask was created by adjusting the threshold intensity to remove the background. Only complete cells including the cell body and cell processes were considered for the analysis. The voxel value was set up at 0.092 according to the resolution of the images. To determine cell geometry including the complex tree topology and the gross spatial arrangement, we analyzed each condition using NeuronStudio 0.9.92 software (Computational Neurobiology and Imaging Center, Mount Sinai School of Medicine, New York) (Rodriguez et al, 2006). Briefly, by defining the localization of the soma, the program automated the collection of the data for cell morphology including the process length, cell volume, and number of branch points and intersections. The results were exported into an Excel spreadsheet. Graphical and statistical analyses comparing all four conditions were done using GraphPad Prism v5.

## Western blot

For Western blot (WB) analyses, cultured NSPCs or hippocampal tissues (at least n = 3 per condition) were collected at indicated time points for each experiment with RIPA buffer (Cat#A32963; Thermo Fisher Scientific) plus a protease inhibitor cocktail with the addition of phosphatase inhibitors (Cat#78441; Thermo Fisher Scientific). For cultured NSPCs, 400,000 cells per condition were used for cell signal transduction purposes. To study the effects of EGFR inhibition

or zinc stimulation, cells were rinsed and cultured with starving media without growth factors during 2 h to down-regulate EGFR downstream signaling. Then, cells were either stimulated with 200 $\mu$M of zinc, 100 ng/ml of HB-EGF (SRP6050-10UG; Sigma-Aldrich), or 20 ng/ml of EGF or pretreated with 2 $\mu$M gefitinib for 1 h. After homogenization, samples were centrifuged at 21,920$g$ for 10 min. The supernatant protein (20 $\mu$g) was loaded in 10% Tris–glycine gels and transferred to a nitrocellulose membrane (Cat#88018; Life Technologies). Blots were blocked in 5% nonfat dry milk in TBS-T (150 mM NaCl, 20 mM Tris–HCl, pH 7.5, 0.05% Tween-20) and then incubated with the following antibodies: phospho-specific antibodies against EGFR (Tyr1068, 1:1,000, #3777; Tyr845, 1:1,000, #2231), AKT (Ser473, 1:1,000, #9271), ERK (Thr202/Tyr204, 1:2,000, #4370), and STAT3 (Tyr705) (1:1,000, #9131; Cell Signaling); total EGFR (1:1,000, ab52894; Abcam); AKT (1:1,000, #9272), ERK (1:1,000 #4695), STAT3 (1:2,000, #4904), and FGFR (1:1,000, #9740) from Cell Signaling; and MTH1/2 (1:1,000, MA1-25479; Thermo Fisher Scientific). Anti-ß-actin (1:1,000, #3700; Cell Signaling) was used for a loading control, and Ponceau staining (P7170; Sigma-Aldrich) was carried out to check protein and band migration. After three washes in TBS-T, blots were incubated with 1:1,000 of either anti-mouse or anti-rabbit IgG HRP-conjugated antibodies (Life Technologies) and developed by an ECL SuperSignal (#34095; Thermo Fisher Scientific) WB analysis system. ECL signal was captured using ChemiDoc Imaging System (Bio-Rad). Quantification of the signal was performed by densitometric scanning of the membrane using Gel-PRO analyzer software (Media Cybernetics, 1993-97).

### Enzyme-linked immunosorbent assay

The HB-EGF content was determined in duplicates by R&D DuoSet Immunoassay (R&D) as described previously (Canals et al, 2004). Briefly, either cellular pellet or brain tissue was sonicated using a buffer containing Hepes 25 mM (Cat#54457; Sigma-Aldrich), MgCl2 5 mM (Cat#M4880; Sigma-Aldrich), ethylene glycol-bis($\beta$-amino-ethyl ether)-N,N,N′,N′-tetraacetic acid (EGTA; Cat#E3889; Sigma-Aldrich) 1 mM, EDTA (Cat#E6758; Sigma-Aldrich) 1.3 mM, PMSF (Cat#PMSF-RO; Sigma-Aldrich) 1 mM, and a protease and phosphatase cocktail inhibitors (Ref. 88,665 and 78,420, respectively; Thermo Fisher Scientific). For cultured NSPCs, 250,000 cells were seeded in a 96-well plate with 150 $\mu$l of culture media without growth factors, and the cell-containing media were collected after 72 h of plating. We analyzed 100 $\mu$l of both the supernatant and lysed cell pellet from the NSPC culture, diluting them 1:1 with reagent buffer 1X. In parallel conditions, right after 72 h of culture, the number of cells was counted using a Bio-Rad TC20 automated cell counter (Bio-Rad Laboratories), using trypan blue (Cat#302643; Sigma-Aldrich) to mark and exclude death cells. Next, we determined the amount of HB-EGF per cell through ELISA (Cat#DY8239-05; R&D Systems). For analysis of HB-EGF levels in the brain tissue, mice were deeply anesthetized in a CO2 chamber at 1.5, 12, 24, and 72 h post-KA or saline administration (n = 4). Their hippocampi were removed, and the DG was dissected out on ice and rapidly frozen using $CO_2$ pellets. Samples were homogenized in the above-mentioned lysis buffer and centrifuged for 20 min at 14,000 rpm at 4°C. Supernatants were collected, and the total protein content was analyzed using a basic colorimetric assay protein assay kit (23227; Pierce). One hundred micrograms of protein was loaded for each

time point and diluted 1:1 in reagent diluent. Both in vitro and in vivo quantifications were performed using the following four-parameter logistic (4-PL) curve fit: y = d+(a-d)/(1+(x/c)^b), where x and y were the independent and dependent variables, respectively, a and d were the minimum and maximum values measured, c was the point of inflection (halfway between a and d), and b was Hill's slope of the curve (steepness of the curve at point c). In vitro values were normalized and expressed as the picogram of HB-EGF per mL, and in vivo values were calculated as the pg of HB-EGF per $\mu$g of the tissue protein.

### Danscher's staining

The indirect autometallographic Danscher method is based on the precipitation of metal ions as insoluble salts that can be visualized by microscopy. Twenty milligrams/kilograms of sodium selenite (S5261-10G; Sigma-Aldrich) was injected intraperitoneally in n = 3 control and n = 3 KA Nestin-GFP animals 30 min before perfusion with a PFA 4%/glutaraldehyde 0.5% (Cat#16536-05; Electron Microscopy Sciences) mixture. Brains were dissected, and serial vibratome sections were made (50 $\mu$m thick) using a Leica VT 1200S vibrating blade microtome (Leica Microsystems GmbH). Autometallography of zinc–selenium nanocrystals was developed as previously described (López-García et al, 2002). The granule size quantification was carried out by determining the region of interest (ROI), in this case, GFP-expressing cells (15–20 cells per condition), and using the particle analyzer of ImageJ software.

### Image capture

Immunofluorescence images were collected by employing the 20X or the 40X oil-immersion objective of a Leica SP8 (Leica) laser scanning microscope and their corresponding manufacturer's software (LAS X; Leica Microsystems). For the quantification of total areas of the DG, the 10X objective was used to completely visualize the DG in each section. The zinc granules in Fig 4 were quantified using an Olympus BX31 microscope with an attached Olympus DP72 high-sensitivity camera (Olympus). The brightfield (light) microscopy was used for zinc granules and was overlapped with the green fluorescence channel was used for Nestin-GFP signal. The signal from each fluorochrome was collected sequentially, and controls with sections stained with the fluorochrome were performed to confirm the absence of signal leaking into different channels. Brightness, contrast, and background were adjusted equally for the entire image without any further modifications. All images shown are flat projections from z-stacks ranging from 10 (typically for individual cell images) to 20 $\mu$m of thickness. 4–5 z-stacks located at random positions in the DG were collected per hippocampal section, and 4–6 sections per series were analyzed, depending on the experiment. For cultured NSPCs, five pictures of 4-$\mu$m-thick random z-stacks of the coverslips were collected per sample and condition.

### Cell quantification

Quantitative analysis of cell populations in vivo was performed by design-based (assumption-free, unbiased) stereology using a

modified optical fractionator sampling scheme as previously described (Encinas et al, 2006, 2011; Encinas & Enikolopov, 2008). For cell densities, quantifications were done by maintaining the same z-stack size and settings among conditions. The values were normalized to the volume of the SGZ+GCL (granule cell layer). For total numbers, the whole area of the DG was determined in every slice of a series and then multiplied by the thickness of the GCL, which was measured at least at three different points in each brain slice to obtain an average. The obtained value was multiplied by the number of series obtained during vibratome sectioning (five or six) to obtain the volume of the whole DG.

NSCs were counted following the previously described criteria for their identification (Kronenberg et al, 2003; Encinas & Enikolopov, 2008), defining them as radial glia-like cells positive for Nestin-GFP and GFAP with the soma located in the SGZ or the lower third of the GCL and with a process extending from the SGZ toward the molecular layer through the GCL. ANPs were defined as Nestin-GFP–positive cells devoid of GFAP immunostaining and with none or short horizontal processes. The cell cycle marker Ki67 or administration of BrdU and its posterior immunolabeling were used to identify proliferating cells. Proliferation of cultured NSPCs was measured by the incorporation of BrdU. The relative proportions of BrdU-positive NSPCs were referred to the total number of cells (DAPI staining) quantified per coverslip and condition. The proliferation in all the conditions was referred to as the basal proliferation of the control condition.

To measure cell death, apoptotic or necrotic cells were defined as cells with small condensed nuclei with abnormal morphology (pyknotic/karyorrhectic). GCD (distance from the SGZ to the molecular layer) was measured in DAPI- or NeuN-stained sections, and the thickness of the GCL was measured at least at three points in each slice.

For the analysis of EGFR, an automated batch analysis for ImageJ was used. In brief, the area was measured as the pixels occupied by EGFR. The ROI was selected in the Nestin-GFP channel using the tool "Color threshold, HSB color space" to adjust the ROI for NSCs. The generated mask was overlaid as the ROI in the rest of channels. Automated quantification was obtained with the "find maxima" option, using the same "noise-tolerance" for all conditions, obtaining the total area occupied by EGFR. For the analysis of EGFR expression in cell culture, at least 25 mitotic or interphasic cells were analyzed. Using the tool "polygon" of ImageJ, the outline of the cells in the brightfield channel was delimited to determine cell area and create the respective ROI for each cell, representing cell area and the ratio of pixel intensity of EGFR signal per cell area.

We analyzed the whole dentate gyrus for absolute numbers of cells (Ki67, dead cells…) and the medial half of the hippocampus for analyses related to NSCs and React-NSCs. As the dentate gyrus changes its orientation so do NSCs, and it is more difficult to identify them in the most lateral aspect of the hippocampus.

### Statistical analysis

The statistical testing was implemented using Sigma-Aldrich Plot version 12 (Systat Software), except from the Sholl analysis in Fig 3F–H in which GraphPad Prism version 5 was used. Comparisons between multiple groups were made using one-way ANOVA or the Kruskal–Wallis test when the data did not comply with normality. The analyses were followed by multiple comparisons: all pairwise in Figs 3E–H, 5C–K, S3G, S4F, and S5C; versus the control group in Figs 1B–G and S4A–D. The Holm–Sidak post hoc method was used for multiple comparisons after one-way ANOVA in all cases except for Fig 5F, where the Fisher LSD method was performed. The Kruskal–Wallis test was followed by post hoc Dunnett's method in Figs 1E and F and S4A for MT1; by post hoc Dunn's method in Fig S4D–F; and by the post hoc Student–Newman–Keuls method in Figs 5C and H and S3B and G and S5C. Comparisons between only two groups were made using a two-tailed unpaired $t$ test or a Mann–Whitney U test when the data did not comply with the normality assumption. All statistical tests and the number of independent experiments can be found in the figure legends. $P$ 0.05 was considered as statistically significant. Results were presented as the mean ± SEM.

## Data Availability

The data that support the findings of this study are available from the corresponding authors upon reasonable request.

### Ethics statement

We confirm that we have read the Journal's position on issues involved in ethical publication and confirm that this report is consistent with those guidelines.

## Supplementary Information

## Acknowledgements

We thank the staff at the Leioa animal facility of UPV/EHU, Laura Escobar at the Imaging Core Facility in Achucarro, and the whole Laboratory of Neural Stem Cells and Neurogenesis Laboratory for insight and discussion. This work has been funded by the Spanish Ministry of Economy and Competitiveness (MINECO, with FEDER Funds) grants SAF2-015-70866-R and MCIN/AEI/10.13039/50110001103 PID2019-104766RB-C21 to JM Encinas-Pérez and JR Pineda; by the MINECO Ramón y Cajal Program: RYC-2013-13450 to JR Pineda and RYC 2012-11137 to JM Encinas-Pérez; and by the MINECO PCIN-2016-128 (ERA-NET-NEURON III program) to JM Encinas-Pérez; O Pastor-Alonso held a UPV/EHU predoctoral fellowship, I Durá held an FPI (MINECO) predoctoral grant, and S Martín-Suárez held a Fundación Gangoiti predoctoral fellowship.

### Author Contributions

O Pastor-Alonso: conceptualization, data curation, formal analysis, validation, investigation, visualization, methodology, and writing—original draft, review, and editing.

I Durá: formal analysis, investigation, and methodology.
S Bernardo-Castro: formal analysis, investigation, and methodology.
E Varea: investigation, visualization, and methodology.
T Muro-García: formal analysis, investigation, and methodology.
S Martín-Suárez: formal analysis, investigation, and methodology.
JM Encinas-Pérez: conceptualization, resources, formal analysis, supervision, funding acquisition, validation, investigation, visualization, project administration, and writing—original draft, review, and editing.
JR Pineda: conceptualization, data curation, formal analysis, supervision, funding acquisition, validation, investigation, visualization, methodology, and writing—original draft, review, and editing.

## Conflict of Interest Statement

The authors declare that they have no conflict of interest.

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
