## [Reviewer comments · Life Science Alliance]

Life Science Alliance

HB-EGF activate EGFR to induce reactive neural stem cells in the mouse hippocampus after seizures

Oier Pastor-Alonso, Irene Dura, Sara Bernardo-Castro, Emilio Varea, Teresa Muro-Garcia, Soraya Martin-Suarez, Juan Encinas-Perez, and Jose R. Pineda

DOI: <https://doi.org/10.26508/lsa.202201840>

Corresponding author(s): Juan Encinas-Perez, Achucarro Basque Center for Neuroscience and Jose R. Pineda,

Review Timeline:

Submission Date:	2022-11-18
Editorial Decision:	2023-01-13
Revision Received:	2024-05-20
Editorial Decision:	2024-06-10
Revision Received:	2024-06-18
Accepted:	2024-06-20

Transaction Report:

January 13, 2023

Re: Life Science Alliance manuscript #LSA-2022-01840-T

Juan Manuel Encinas-Perez
Achucarro Basque Center for Neuroscience
Spain

Dear Dr. Encinas-Perez,

Thank you for submitting your manuscript entitled "HB-EGF and zinc activate EGFR to induce reactive neural stem cells in the mouse hippocampus after seizures" to Life Science Alliance. The manuscript was assessed by expert reviewers, whose comments are appended to this letter. We invite you to submit a revised manuscript addressing the Reviewer comments.

Thank you for this interesting contribution to Life Science Alliance. We are looking forward to receiving your revised manuscript.

Sincerely,

B. MANUSCRIPT ORGANIZATION AND FORMATTING:

Reviewer #1 (Comments to the Authors (Required)):

The manuscript by Alonso and colleagues uses an acute mouse model of temporal lobe epilepsy (MTLE) to investigate the mechanisms underlying the activation of hippocampal neural stem cells to a reactive state, with the concomitant impairment of adult neurogenesis that emerges after seizures. The authors delivered kainic acid (KA) to induce seizures and showed that this treatment increased the expression and phosphorylated status of EGFR in NSCs, 1-3 days after seizure induction. It is then shown that Gefitinib (Gef), an EGFR blocker, impairs NSC proliferation in culture. In vivo experiments also show that Gef reduces the expression of Ki67 in the SGZ after KA, an indication of diminished proliferation. KA treatment also depletes neurogenesis, as shown by a decrease in DCX+ cells after 14 days. This effect was blocked by Gef, involving the activation of EGFR in the effects of KA. The manuscript shows a clear connection between the effects of KA and the requirement of EGF signaling. However, the following sections of the manuscript are less connected to the initial phenomenon. It is shown that KA produces a marginal increase in HB-EGF, a natural ligand of EGFR, and it also increases Zn levels. While the manuscript attempts to link KA, EGFR, and Zn, this connection is not entirely convincing. The main concerns are described below.

Major critiques:

- 1) While Fig. 3C shows an experiment where BrdU is injected after KA in the presence or absence of Gef, the result is not shown. There is a missing figure that should show precisely the density of BrdU+ cells with KA or with KA+Gef (not as normalized proportions, but actual cell density as shown in Fig. 4E, for instance). This is important to then compare the KA effects with those induced by Zn²⁺ treatment. This analysis will help to determine how similar or different are those conditions.
- 2) KA treatment increases HB-EGF, a natural ligand of EGFR. The authors find an increase from about 800 to 1000 pg/100mg tissue, about 25% (Fig. S3). It is unlikely that this limited increase explains the striking effect of KA in NSCs, cell proliferation or neurogenesis. Therefore, the argument that this ligand is relevant to the effect of KA is weak.
- 3) KA 72h increases the levels of Zn (Fig. 4A,B). The Danscher staining used to arrive to this conclusion is not convincing, it does not show positive or negative controls (a good positive control would be an intracerebral Zn injection). In addition, the extent of increase would not seem to justify the striking effects of KA.
- 4) To compare the extent of the process induced by Zn 20-30 nmol to the one induced by KA, BrdU cells should be counted in the same manner. The equivalent to Fig. 4E for KA is not shown (cell density).
- 5) The level of GCL dispersion shown in the examples (Fig. 4D) is inconsistent with the quantified data (Fig. 4F), which shows a 20% change.
- 6) Effects of Zn in culture are intriguing. Fig. S4D shows that 2.5 μ m increases proliferation, but 10 μ m abolishes it completely. The contribution of these experiments is unclear.
- 7) It is stated that Zn²⁺ disorganizes the GCL in a similar manner as KA. However, this direct comparison has not been done quantitatively (the KA equivalent for Fig. 4F is missing).

Minor comments:

Fig. 3G: wrong position of the mean bar in KA+DMSO.

Fig. 4: panel C is cut on the bottom. Two panels are labeled "E".

Reviewer #2 (Comments to the Authors (Required)):

This manuscript by Pastor-Alonso and colleagues describes the role of epidermal growth factor receptor (EGFR) signaling in the development of reactive hippocampal neural stem cells (NSCs) and impaired adult neurogenesis after intrahippocampal KA (IH-

KA) injection. They also propose a role for HB-EGF and zinc in this signaling pathway. Overall, this manuscript presents some interesting observations into the role of EGFR in impaired adult neurogenesis after IH-KA. Enthusiasm is tempered, however, by a lack of data supporting some of the conclusions, and other data described in the text that is not present in the figures. In addition, many of the figure panels are referred to incorrectly. The main concerns include the following:

1. In Figure 1H, are some of the double labeled cells in the hilus reactive astrocytes rather than NSCs? Is GFP expressed by reactive astrocytes in this Nestin-GFP reporter line?
2. The data in Suppl Fig. 2A, B are difficult to interpret without a mitosis-specific marker.
3. In Fig. 3B, please describe for the reader how NSCs and ANPs were identified.
4. The authors refer to Fig. 3A, B for gefitinib reducing granule cell dispersion after in vivo KA, but I cannot find these data. If anything, the KA+Gefitinib granule cell layer appears nearly as dispersed as KA+DMSO in Fig. 3D.
5. The authors describe early reduction in cell proliferation in vivo at 72 hours post-KA after gefitinib treatment and refer to Fig. S3A, C, but no such data are present in the figure.
6. Page 11 top, the authors state that "Although diminished BrdU+ ANPs and total BrdU cells were observed, they did not reach significance, suggesting a more specific effect of EGFR blocking in NSCs in terms of proliferation." This conclusion conflicts with the Ki67 data in Fig. 3A, B.
7. It is not clear why day 7 was used for the Zinc injection experiments when that time point was not used for any other experiments, making comparisons impossible.
8. Page 14, the authors conclude that endogenous zinc exerts a neuroprotective effect. It is not clear what data this was based upon given that zinc chelation did not increase cell death after KA (Fig. S4F).
9. Page 16, the authors state that "Our results showed that indeed EGFR inhibition reduced the increase of BrdU+ cells in the GCL provoked by the intrahippocampal injection of zinc". However, most of the BRDU+ cells appear outside the GCL in Fig. 5J.
10. The data referring to DCX+ cell numbers after Zinc injection (Page 16) are not in any of the figures.
11. The authors claim that they show that zinc is involved in the EGFR-dependent induction of React-NSCs, but this conclusion is not supported by the data (e.g., Zinc chelation did not block any of the effects of IH-KA).
12. The authors also conclude that gefitinib blocked the effects of zinc on cultured NSPCs in vitro and in the neurogenic niche in vivo. Other than p-ERK changes, however, these effects were quite modest or absent.

Minor issues:

13. First paragraph of Intro: Surgery is no longer considered "as a last-resort therapeutic strategy". It should be considered after a patient fails 2 good trials of anti-seizure medications.
14. The panels in Fig. S3 are referred to incorrectly in the text.
15. Page 13 top, the text refers to 5 μ M zinc, but Fig. S4C, D describe 2.5 μ M zinc. Also, it is difficult to see the clear loss of cytoplasmic Nestin-GFP, as well as the abundant presence of bright and small pycnotic nuclei in Fig. S4C.
16. Fig. 4 panel D is mislabeled as E.
17. The authors describe "a 37% reduction of BrdU-positive cells in presence of gefitinib" in Fig. 2E - do they mean a 63% reduction?
18. Grammatical issues should be corrected - for example, "signalization" should be replaced by "signaling" and "hippocampalectomy" by "amygdalo-hippocampectomy" (no surgeons solely remove hippocampus).

Reviewer #3 (Comments to the Authors (Required)):

This manuscript by Pastor-Alonso et al. addresses the signaling pathways which contribute to the distortion of the neurogenic niche in the hippocampus upon seizures that resemble to some extent the pathology of mesial temporal lobe epilepsy (MTLE). By applying in vitro assays, biochemistry and in vivo studies the authors identify the EGFR pathway and Zinc release as cause for reactive NSCs, the generation of reactive astrocytes and the granule cell dispersion in the DG. Importantly administration of

the inhibitor gefitinib prevents numerous effects of kainic acid induced MTLE.

This work is very well conducted and adds important new insight into the pathways that cause the detrimental effects on the adult neurogenic lineage that can further potentiate pathology in epilepsy. Prior to publication, it would be beneficial to clarify certain aspects in the manuscript.

Which cell type is upregulating EGFR expression in the hilus upon KA application (Fig. 1H) and could this cell type potentially add to the observed effects on proliferation and differentiation in the SGZ?

Does Gefitinib have a direct effect on the expression of EGFR? The expression seems to be increased upon treatment (Fig. 2A).

Does Gefitinib reverse or prevent GCD? Further I do not see evidence for GCD reverse in the referred Fig. 3A,B as this is the quantification of proliferation, nor in Fig. 3 D.

Which cell type within the DG is supposed to release HB-EGF? Is there experimental indication of knowledge from the literature?

Why are no radial Nestin+ processes visible in Fig. 4A? Another field of view with a clearly detectable GFP+ process would be helpful for comparison.

p14: "However, TPEN failed to reduce cell death, as a similar density of pycnotic nuclei was found in the GCL than KA+vehicle animals and significantly higher than saline+TPEN animals (Supplementary Figure 4F). These results suggest that despite mimicking the early effects of MTLE-HS on the neurogenic niche, endogenous zinc also exerts a neuroprotective effect." If Zinc has a neuroprotective effect, wouldn't one expect to see higher levels of pyknotic nuclei in Fig. 4F KA+TPEN case?

In general, there are several wording, grammar and typ-o mistakes throughout the manuscript text which would need to be corrected.

Reviewer #1 (Comments to the Authors (Required)):

The manuscript by Alonso and colleagues uses an acute mouse model of temporal lobe epilepsy (MTLE) to investigate the mechanisms underlying the activation of hippocampal neural stem cells to a reactive state, with the concomitant impairment of adult neurogenesis that emerges after seizures. The authors delivered kainic acid (KA) to induce seizures and showed that this treatment increased the expression and phosphorylated status of EGFR in NSCs, 1-3 days after seizure induction. It is then shown that Gefitinib (Gef), an EGFR blocker, impairs NSC proliferation in culture. In vivo experiments also show that Gef reduces the expression of Ki67 in the SGZ after KA, an indication of diminished proliferation. KA treatment also depletes neurogenesis, as shown by a decrease in DCX⁺ cells after 14 days. This effect was blocked by Gef, involving the activation of EGFR in the effects of KA. The manuscript shows a clear connection between the effects of KA and the requirement of EGF signaling. However, the following sections of the manuscript are less connected to the initial phenomenon. It is shown that KA produces a marginal increase in HB-EGF, a natural ligand of EGFR, and it also increases Zn levels. While the manuscript attempts to link KA, EGFR, and Zn, this connection is not entirely convincing. The main concerns are described below.

We thank the reviewer for the positive comments and for the time and effort dedicated to improve the manuscript. We have addressed all the points of concern. Changes in the text are written in red for better tracking.

Major critiques:

1) While Fig. 3C shows an experiment where BrdU is injected after KA in the presence or absence of Gef, the result is not shown. There is a missing figure that should show precisely the density of BrdU⁺ cells with KA or with KA+Gef (not as normalized proportions, but actual cell density as shown in Fig. 4E, for instance). This is important to then compare the KA effects with those induced by Zn²⁺ treatment. This analysis will help to determine how similar or different are those conditions.

We apologize for the confusion caused by the BrdU missing figure. We agree with the reviewer that BrdU⁺ cells/mm³ could give an idea about how comparable are both phenotypes. We have done the corresponding new experiment. Density of BrdU⁺ cells from the end of the treatment up to 14 days post KA are now presented in Supplementary Figure 3.

We added the text in page 11 of the manuscript: “At the end of treatment, we administered three pulses of BrdU intraperitoneally, separated by 3h, to determine the density of newly-generated cells produced once the inhibition of proliferation has ceased. We observed a rebound trend in the saline+gefitinib (16.330±2550 BrdU⁺ cells), saline+KA and KA+gefitinib (12.950±2430 and 14.190±5930 cells) groups and found no statistically significant difference respect to the saline+DMSO group (4940±3470 cells; One-way ANOVA p=0.256; Supplementary Figure 3C).”

It should be considered that data on Fig 3C refers to 14 days post KA injection, meanwhile data of Fig.4E refers to 7 days post Zn injection, making these data not directly comparable. We added a 7-day KA injection time point to Figure 4 to compare both models. However, the scope of the manuscript is to highlight that Zn²⁺ release that occurs in KA epileptic model is just one of the actors that help contribute to KA phenotype, but not the sole mechanism. Indeed, not only inflammation and cell death (as we state in Supplementary Fig 4E and 4F (TPEN)) but also recurrent cell hyperactivation can contribute to the massive changes of the DG observed in KA phenotype that could increase BrdU-positive cells.

2) KA treatment increases HB-EGF, a natural ligand of EGFR. The authors find an increase from about 800

to 1000 pg/100mg tissue, about 25% (Fig. S3). It is unlikely that this limited increase explains the striking effect of KA in NSCs, cell proliferation or neurogenesis. Therefore, the argument that this ligand is relevant to the effect of KA is weak.

We thank the referee for bringing up this important point. We never intended to claim that HB-EGF is the unique mechanism responsible of KA effects in NSCs, cell proliferation or neurogenesis. The increment in HB-EGF alone might not be sufficient but the increase in EGFR has to be considered also. As any complex biological process such as reactive gliosis and neuroinflammation, the induction of React-NSCs is likely to be a multifactorial one. In any case, changes of HB-EGF expression in the hippocampus were already reported as very early events in the KA model (Opanashuk LA et al. 1999 *J Neurosci.* 19(1): 133–146) but no further investigation followed and the relationship with neurogenesis or NSCs has not been addressed so far. In 1999 Opanashuk interpreted that the role of HB-EGF was excitoprotective. 24 year later, we propose that this growth factor plays an important role in the dramatic changes underwent by NSCs and neurogenesis in experimental MTLE.

3) KA 72h increases the levels of Zn (Fig. 4A,B). The Danscher staining used to arrive to this conclusion is not convincing, it does not show positive or negative controls (a good positive control would be an intracerebral Zn injection). In addition, the extent of increase would not seem to justify the striking effects of KA.

We thank the reviewer for its suggestion and we have done more experiments and provided further information. The results are shown in Supplementary Figure 4.

“To corroborate the findings, we did an immunofluorescence against zinc transporter (ZnT3) which takes up Zinc ions into synaptic vesicles (Wenzel et al., 1997; Cole et al., 1999) after 3dPKA. We observed a dramatic increase of ZnT3 pixel staining intensity at granule cell layer from 14997.50 ± 4041.08 to 60072.00 ± 7985.31 for saline and KA-injected, ZnT3 pixel staining intensity overlapping Nestin+GFP increased from 9462.90 ± 3857.63 to 22628.57 ± 7620.12 and Hilus region from 76090.87 ± 20042.37 to 195283.71 ± 28167.48 for saline and KA-injected respectively (Student’s t-test $p < 0.001$, $p < 0.05$ and $p < 0.01$; Supplementary Figure 4A-B) in agreement with Danscher data.”

According to literature the addition of zinc to reveal already dense zinc staining has not been found and probably could be due to the masking differences on already black dotted staining. An alternative way (apart from 20 years of literature) to highlight the presence of Zinc in KA model could rely in the zinc transporter “ZnT3” which takes up Zinc ions into synaptic vesicles (Wenzel HJ. et al., 1997; Cole TB. et al., 1999). Immunostaining against ZnT3 shows a dramatic increase of ZnT3 after 3dPKA in agreement with Danscher data. Furthermore, it is already known that ZnT3^{-/-} mice in response to kainic acid presents seizure-related neuronal damage (Cole TB., et al 1999) as synaptically released zinc has neuromodulatory capabilities that could result in either inhibition or enhancement of neuronal excitability. We added ZnT3 staining in Supplementary Figure 4A-B (see image below) and the following text in page 12:

Although we acknowledge the shortcomings of the techniques we would like to remind that the Timm or Danscher staining is a technique used since 1981 to visualize zinc-containing neurons and the detection of newly sprouted axons and axon terminals within the central nervous system. Routbort MJ and collaborators already found in 1999 (Routbort MJ et al 1999) using Timm stain already found dense granular staining in the dentate hilus and a smaller amount of mossy fiber-like staining in the inner molecular layer. Indeed, Treatment of slice cultures with KA for 48 h produced robust mossy fiber sprouting several weeks later as gauged by Timm staining. Also, Paul S. Buckmaster and Edward Dudek found in KA-treated rats larger percentages of Timm staining in the granule cell layer plus molecular layer compared with control rats (Buckmaster PS and Dudek FE 1997).

References:

Buckmaster, P.S. and Dudek, F.E. (1997), Neuron loss, granule cell axon reorganization, and functional changes in the dentate gyrus of epileptic kainate-treated rats. *J. Comp. Neurol.*, 385: 385-404. [https://doi.org/10.1002/\(SICI\)1096-9861\(19970901\)385:3<385::AID-CNE4>3.0.CO;2-#](https://doi.org/10.1002/(SICI)1096-9861(19970901)385:3<385::AID-CNE4>3.0.CO;2-#)

Cole TB, Wenzel HJ, Kafer KE, Schwartzkroin PA, Palmiter RD (1999) Elimination of zinc from synaptic vesicles in the intact mouse brain by disruption of the ZnT3 gene. *Proc Natl Acad Sci USA* 96: 1716-1721.

M.J Routbort, S.B Bausch, J.O McNamara, M. J. Routbort and S. B. Bausch. (1999) Seizures, cell death, and mossy fiber sprouting in kainic acid-treated organotypic hippocampal cultures. *Neuroscience*, Volume 94, Issue 3, Pages 755-765,

Wenzel HJ, Cole TB, Born DE, Schwartzkroin PA, Palmiter RD (1997) Ultrastructural localization of zinc transporter-3 (ZnT-3) to synaptic vesicle membranes within mossy fiber boutons in the hippocampus of mouse and monkey. *Proc Natl Acad Sci USA* 94: 12676-12681.

4) To compare the extent of the process induced by Zn 20-30 nmol to the one induced by KA, BrdU cells should be counted in the same manner. The equivalent to Fig. 4E for KA is not shown (cell density).

Thank you for this useful remark. In the new Figure 4D now we provide a comparative image of the disorganization induced by 1nmol of KA 7 days post-intrahippocampal KA. Furthermore, we provide BrdU quantification in panel E. The new quantifications are done with the 20 nmol dose only because of the lethality induced by the 30 nmol. To avoid possible confusions, we specify in the legend figure that the quantification corresponds to the dose of 20nmol.

5) The level of GCL dispersion shown in the examples (Fig. 4D) is inconsistent with the quantified data (Fig. 4F), which shows a 20% change.

The quantification provided in Fig 4F is referred to the dispersion of Fig4D corresponding to the dose of 20 nmol of Zn. To avoid possible confusions, we modified accordingly the text of the figure legend: “C, Schematic of the in vivo zinc intrahippocampal injection and BrdU administration. Nestin-GFP mice were sacrificed 7 days after zinc or KA 1nM administration (20nmol) and BrdU was given 24h prior to sacrifice to identify mitotic cells. D, Confocal microscopy images after immunostaining for GFP, BrdU and GFAP. DAPI was used as nuclear staining. Increased cell proliferation, GCD and induction of React-NSCs are noticeable. Scale bar 10 μ m. E, Quantification of the density of BrdU+ cells in the SGZ+GCL. (F) Quantification of GCD. One-way ANOVA * $p < 0.05$ and *** $p < 0.001$. Dots show individual cells (E, F) $n=3-4$.”

6) Effects of Zn in culture are intriguing. Fig. S4D shows that 2.5 μ m increases proliferation, but 10 μ m abolishes it completely. The contribution of these experiments is unclear.

Depending on its level, Zn it can be anything from proliferation-stimulating to lethally toxic. Indeed, Zn is essential for cell proliferation including the regulation of DNA synthesis and mitosis (See review Beyersmann & Haase 2001) and its removal from the extracellular milieu results in the decreased activity of deoxythymidine kinase and also reduced levels of adenosine(5') tetraphosphate(5')-adenosine. Hence, Zn²⁺ may directly regulate DNA synthesis through these systems (MacDonald RS 2000). However, excessive amount of Zn can be toxic through the production of toxic reactive oxygen species (ROS) and the inhibition of glutathione reductase in astrocytes (Bishop GM et al 2007).

Both effects on proliferation and death can be found in our experiments although we did not investigate further these observations because they fall out of the scope of the manuscript. We agree that the contribution of this experiment does not bring anything new and to clarify the message we decided to remove it from the manuscript.

References:

D Beyersmann, H Haase. Functions of zinc in signaling, proliferation and differentiation of mammalian cells. *Biometals*. 2001 Sep-Dec;14(3-4):331-41. doi: 10.1023/a:1012905406548.

Glenda M Bishop, Ralf Dringen, Stephen R Robinson. Zinc stimulates the production of toxic reactive oxygen species (ROS) and inhibits glutathione reductase in astrocytes. *Free Radic Biol Med*. 2007 Apr 15;42(8):1222-30. doi: 10.1016/j.freeradbiomed.2007.01.022.

R S MacDonald. The role of zinc in growth and cell proliferation. *J Nutr*. 2000 May;130(5S Suppl):1500S-8S. doi: 10.1093/jn/130.5.1500S.

7) It is stated that Zn²⁺ disorganizes the GCL in a similar manner as KA. However, this direct comparison has not been done quantitatively (the KA equivalent for Fig. 4F is missing).

We thank to the reviewer for this useful comment. We showed KA dispersion at 14 days and it is already known that KA induces GCL dispersion gradually over time. We did another experiment with perfusion and analysis 7 days after the KA injection for direct comparison. The images and the GCL measures of saline, Zn²⁺ 20 nmol and KA 1nmol has been added to the Fig 4D and F panels. We added the following text in page 13: "A clear induction of React-NSCs was observed and a significant induction of GCD was measured: 70.96±3.14 μm (saline), 85.75±1.78 μm 20nmol Zn²⁺ and 124.210±4.67 μm 1nmol KA (One-way ANOVA p<0.001; Figure 4F)." As mentioned before, due to the high mortality induced by the dose of Zn²⁺ 30 nmol., we did not use this dose for quantifications.

Minor comments:

Fig. 3G: wrong position of the mean bar in KA+DMSO.

Thank you for the attention to detail. Indeed, the mean bar is correct but the high dispersion of the dots for KA conditions made us to shrink the interval of presented values to a homogenous range (between 0 and 800). In this way was easier to appreciate small variations. However, in agreement with the reviewer we provide the complete representation of the plot data considering all the range is as follows and changed accordingly in the new Figure 3G.

Fig. 4: panel C is cut on the bottom. Two panels are labeled "E".

Thank you very much for the remarks. We apologize for the mistakes, that have been corrected in the present version.

Reviewer #2 (Comments to the Authors (Required)):

This manuscript by Pastor-Alonso and colleagues describes the role of epidermal growth factor receptor (EGFR) signaling in the development of reactive hippocampal neural stem cells (NSCs) and impaired adult neurogenesis after intrahippocampal KA (IH-KA) injection. They also propose a role for HB-EGF and zinc in this signaling pathway. Overall, this manuscript presents some interesting observations into the role of EGFR in impaired adult neurogenesis after IH-KA. Enthusiasm is tempered, however, by a lack of data supporting some of the conclusions, and other data described in the text that is not present in the figures. In addition, many of the figure panels are referred to incorrectly. The main concerns include the following:

We appreciate the positive feedback regarding the manuscript and value the comments to improve it. We have addressed all the referee's comments.

1. In Figure 1H, are some of the double labeled cells in the hilus reactive astrocytes rather than NSCs? Is GFP expressed by reactive astrocytes in this Nestin-GFP reporter line?

The Nestin-GFP reporter line labels NSCs but also reactive astrocytes because astrocytes start to express nestin as part of their transformation on reactive astrocytes. In addition, early amplifying neural progenitors (ANPs), oligodendrocyte precursor cells (OPCs) and pericytes are also positive for Nestin-GFP. Morphology and markers are used to distinguish NSCs among these cell types (Encinas and Enikolopov 2008; Encinas et al. 2011; Sierra et al. 2015; Martín-Suárez et al. 2019; Muro-García et al. 2019...). Please take also into account that at the tip of "V" shape of the dentate gyrus both dorsal blade and ventral blade joins. The intention of the image was to find in a sufficiently magnified field of view a quiescent NSC, an activated NSC (with EGFR staining) and a neural progenitor exiting asymmetric cell division. The probability to find these three events in a is quite low and in this case, we found them close the union of the two blades, as can be seen by the appearance of DAPI nuclei corresponding to the ventral region in the lower right part.

See also the response to comment 3.

References:

Encinas JM, Enikolopov G. 2008. Identifying and quantitating neural stem and progenitor cells in the adult brain. *Methods in cell biology* 85:243–72.

Encinas JM, Michurina TV, Peunova N, Park J-H, Tordo J, Peterson DA, Fishell G, Koulakov A, Enikolopov G. 2011. Division-coupled astrocytic differentiation and age-related depletion of neural stem cells in the adult hippocampus. *Cell Stem Cell* 8:566–579.

Sierra A, Martín-Suárez S, Valcarcel-Martín R, Pascual-Brazo J, Aelvoet SA, Abiega O, Deudero JJ, Brewster AL, Bernales I, Anderson AE, Baekelandt V, Maletic-Savatic M, Encinas JM. 2015. Neuronal hyperactivity accelerates depletion of neural stem cells and impairs hippocampal neurogenesis. *Cell Stem Cell* 16:488–503.

Martín-Suárez S, Valero J, Muro-García T, Encinas JM. 2019. Phenotypical and functional heterogeneity of neural stem cells in the aged hippocampus. *Aging Cell*. Aug;18(4):e12958.

Muro-García T, Martín-Suárez S, Espinosa N, Valcárcel-Martín R, Marinas A, Zaldumbide L, Galbarriatu L, Sierra A, Fuentealba P, Encinas JM. 2019. Reactive Disruption of the Hippocampal Neurogenic Niche After Induction of Seizures by Injection of Kainic Acid in the Amygdala. *Front Cell Dev Biol* 7:158.

Valcárcel-Martín R, Martín-Suárez S, Muro-García T, Pastor-Alonso O, Rodríguez de Fonseca F, Estivill-Torrús G, Encinas JM. 2020. Lysophosphatidic Acid Receptor 1 Specifically Labels Seizure-Induced Hippocampal Reactive Neural Stem Cells and Regulates Their Division. *Front Neurosci* 14:811.

2. The data in Suppl Fig. 2A, B are difficult to interpret without a mitosis-specific marker.

We acknowledge the usefulness of this comment. We have added an experiment, in which we used Ki67, a mitotic marker which allows for the visualization of condensed chromosomes at the metaphase plate together with immunolabeling. We also provide a DAPI staining of a telophase without the primary antibodies to corroborate the specificity of the antibodies. We added it to Supplementary Figure 2A. It should be taken into consideration that NSCs and NSPCs cultured grow under proliferating cell culture conditions. Cell roundness and chromosome condensation are hallmarks to identify proliferating cells. We present here in “panel A” a DAPI staining of three nuclei (two interphase and one at the end of anaphase beginning of telophase). The mitotic one is easy to appreciate the chromosomal condensation of daughter cells and it is easy to contrast the different bright intensity of EGFR staining.

3. In Fig. 3B, please describe for the reader how NSCs and ANPs were identified.

We appreciate the importance of this point. NSCs are identified by the co-expression of nestin and GFAP and their unique morphology aspect. From the soma (with nucleus seen with DAPI), located in the SGZ or the most proximal part to the hilus of the GCL, a main apical process nestin and GFAP-positive, extends across the GCL towards the molecular layer. This apical process can be very little ramified as in control NSCs or present more branching as in the React-NSCs of MTL mice. ANPs are located in the SGZ and positive for nestin but not for GFAP. They are rounded and bear a few short prolongations. The explanation (see below) have been added to the main text and the figure legend. See also the references given in comment 1.

“NSCs were identified as being immunopositive for nestin and GFAP and with a radial apical process extending across the GCL from the SGZ to the molecular layer. ANPs are located in the SGZ, are nestin-positive but GFAP-negative and bear few and short process.”

4. The authors refer to Fig. 3A, B for gefitinib reducing granule cell dispersion after in vivo KA, but I cannot find these data. If anything, the KA+Gefitinib granule cell layer appears nearly as dispersed as KA+DMSO in Fig. 3D.

Thank you very much for this detail. We meant “Supplementary Figure 3” (and not Fig.3) when we referred to granule cell dispersion. These data correspond to 14 days post-KA time point. Here we present the graphic related to the text that we added to Supplementary Figure 3.

5. The authors describe early reduction in cell proliferation in vivo at 72 hours post-KA after gefitinib treatment and refer to Fig. S3A, C, but no such data are present in the figure.

We apologize for the mistake. We referred to Fig.3A and B, and not to any supplementary figure.

6. Page 11 top, the authors state that "Although diminished BrdU+ ANPs and total BrdU cells were observed, they did not reach significance, suggesting a more specific effect of EGFR blocking in NSCs in terms of proliferation." This conclusion conflicts with the Ki67 data in Fig. 3A, B.

We thank the reviewer for rising this point that deserves clarification. Overall proliferation (in the SGZ and GCL), and NSC activation, are an early event in the mouse MTLE model. In fact, both peak at 3 days after KA but then diminish. Cell proliferation returns to control levels by day 7 after KA (Sierra et al. 2015). This is the main reason we chose that time point for the experiments corresponding to Fig. 3A and B. Also, KI67 is a marker of mitosis that label cells dividing in the moment of perfusion. In the case of Fig. 3A and B this is 72h after KA. But BrdU administration was done at the end of inhibitory treatments to label the cells that were going to reenter the cell cycle (and label the progeny at 14 days post KA). Also, the ANP direct comparison cannot be done due to difference in the paradigms and potential loss of ANPs by 14 days in KA model. The difference in the experimental design can account for differences in the results.

7. It is not clear why day 7 was used for the Zinc injection experiments when that time point was not used for any other experiments, making comparisons impossible.

We acknowledge the importance of this point. Following also reviewer one's request, we have done new experiments for the MTLE model at 7 days after KA injection. We have added new immunofluorescence labeling images and the corresponding quantification in Fig. 4. Because a single injection of Zn was used the rationale was that the longer we waited for the analysis, the higher the probability of losing a potential effect. Therefore, we compromised by used the 7-day time point, at which the KA injection has marked effects on the neurogenic niche.

8. Page 14, the authors conclude that endogenous zinc exerts a neuroprotective effect. It is not clear what data this was based upon given that zinc chelation did not increase cell death after KA (Fig. S4F).

We acknowledge our lack of precision regarding this statement. Reviewer 3 also commented on this point.

We meant that the lack of Zn due to chelation does not change cell death. Indeed, Zn can be both neuroprotective and neurotoxic (see review Choi S et al 2020). In our case, endogenous Zinc, released from mossy fibers in the hippocampus by seizure activity, is insufficient to alter the extent of neuronal death in agreement with previous work (Lees GC et al 1998).

“However, TPEN failed to alter cell death, as a similar density of pycnotic nuclei was found in the GCL than KA+vehicle animals and significantly higher than saline+TPEN animals (24864.24±630.84 pycnotic nuclei per mm³ for KA+TPEN respect to 2895.61±1635.08 pycnotic nuclei per mm³ for Saline+TPEN; Kruskal-Wallis p=0.014; Supplementary Figure 4F). These results suggest that despite mimicking the early effects of MTLE-HS on the neurogenic niche, endogenous zinc does not contribute to cell death.”

References:

Choi S, Hong DK, Choi BY and Suh SW. Zinc in the Brain: Friend or Foe? *Int. J. Mol. Sci.* 2020, 21(23), 8941; <https://doi.org/10.3390/ijms21238941>

G J Lees, M P Cuajungco, W Leong. Effect of metal chelating agents on the direct and seizure-related neuronal death induced by zinc and kainic acid. *Brain Res.* 1998 Jul 13;799(1):108-17. doi: 10.1016/s0006-8993(98)00483-1.

9. Page 16, the authors state that "Our results showed that indeed EGFR inhibition reduced the increase of BrdU+ cells in the GCL provoked by the intrahippocampal injection of zinc". However, most of the BrdU + cells appear outside the GCL in Fig. 5J.

We appreciate the relevance of this observation. Our work is focused on the neurogenic niche (GCL, including the SGZ) because NSCs and neurogenesis are our main interest and has been the basis of extensive work on the past (see references in answer to comment 1). Other areas in the hippocampus and

interconnected structures are affected by MTLE or its experimental counterpart in terms of cell proliferation, gliosis, neuroinflammation, cell death etc. Many cell types are involved also (microglia, endothelial cells, oligodendrocyte progenitor cells, neurons etc). Characterizing all the responses by all these cell types in the different areas would be a gigantic task with diluted focus and meaning. Therefore, we restricted our study to the neurogenic niche in all the experiments, including those using Zn.

10. The data referring to DCX+ cell numbers after Zinc injection (Page 16) are not in any of the figures.

We are very sorry for the error. We added the data to Supplementary Figure 5 and updated the figure legend accordingly:

11. The authors claim that they show that zinc is involved in the EGFR-dependent induction of React-NSCs, but this conclusion is not supported by the data (e.g., Zinc chelation did not block any of the effects of IH-KA).

We thank the reviewer for bringing up this important point. If Zn chelation would have prevented the induction of React-NSCs and preserved neurogenesis, we could have concluded that Zn was the major player driving the effects of experimental MTLE on the neurogenic niche. This is not our claim. The message we think we can convey from our data is that the EGFR pathway plays an important role (most likely in combination with others) in the induction of React-NSCs and the disruption of neurogenesis associated with experimental MTLE, and that Zn simply contributes to the activation of this pathway. It has been previously reported that endogenous Zn, released from mossy fibers in the hippocampus by seizure activity is insufficient to alter the extent of the resulting neuronal death (Lees GC et al 1998). Also, Zn infusion exerts a mild phenotype of GCD Nestin-GFP cell reactivity over time. We believe that given the bulk of MTLE-induced changes in the neurogenic niche, including gliosis, inflammation and cell death for the *in vivo* approach there is not any single target candidate to revert the observed phenotype, definitely not Zn. We have modified the text accordingly in some places to make sure we convey the right message and acknowledge the shortcomings of the Zn experiments.

Reference:

G J Lees, M P Cuajungco, W Leong. Effect of metal chelating agents on the direct and seizure-related neuronal death induced by zinc and kainic acid. *Brain Res.* 1998 Jul 13;799(1):108-17. doi: 10.1016/s0006-8993(98)00483-1.

12. The authors also conclude that gefitinib blocked the effects of zinc on cultured NSPCs *in vitro* and in the neurogenic niche *in vivo*. Other than p-ERK changes, however, these effects were quite modest or absent.

We acknowledge this point, which is in line with the former comment. We consider that the increase on P-EGFR (845) and its downstream pathway P-ERK is proof that Zn can activate this pathway *in vitro*, and therefore could play a partial role on its direct activation after the KA-injection.

Minor issues:

13. First paragraph of Intro: Surgery is no longer considered "as a last-resort therapeutic strategy". It should be considered after a patient fails 2 good trials of anti-seizure medications.

We appreciate the remark of the reviewer and we have changed the sentence accordingly. We meant that mono- and multi-pharmacological treatments are the first option and then surgery is considered.

“MTLE-HS associates with drug-resistance for anti-seizure medications often leading to unilateral amygdalo-hippocampectomy after failing two trials with anti-seizure medications (Kwan et al., 2009; Crespel et al., 2005; Tatum, 2012).”

And we added the reference:

Patrick Kwan, Michael R. Sperling. (2009) Refractory seizures: Try additional antiepileptic drugs (after two have failed) or go directly to early surgery evaluation? *Epilepsia*. Sep;50 Suppl 8:57-62. doi: 10.1111/j.1528-1167.2009.02237.x.

14. The panels in Fig. S3 are referred to incorrectly in the text.

We apologize for the mistake. We have reworded the sentence accordingly.

15. Page 13 top, the text refers to 5 uM zinc, but Fig. S4C, D describe 2.5 uM zinc. Also, it is difficult to see the clear loss of cytoplasmic Nestin-GFP, as well as the abundant presence of bright and small pycnotic nuclei in Fig. S4C.

We are sorry for the typo error in the text. The purpose of the experiment was to corroborate what is already known in the literature about its dual role on cell proliferation and its toxicity depending of its concentrations. As reviewer one stated that the contribution of this experiments was unclear we have decided to remove it from the manuscript. See response to reviewer one's comment 6.

16. Fig. 4 panel D is mislabeled as E.

We apologize for yet another mistake. We have changed the sentence accordingly.

17. The authors describe "a 37% reduction of BrdU-positive cells in presence of gefitinib" in Fig. 2E - do they mean a 63% reduction?

We thank the reviewer for their attention to detail. Indeed, that is the value of 63% is the one that should be stated. We have corrected the sentence accordingly.

18. Grammatical issues should be corrected - for example, "signalization" should be replaced by "signaling" and "hippocampalectomy" by "amygdalo-hippocampectomy" (no surgeons solely remove hippocampus).

We thank the reviewer for the insight. We have addressed the grammatical and orthographical errors throughout the text.

Reviewer #3 (Comments to the Authors (Required)):

This manuscript by Pastor-Alonso et al. addresses the signaling pathways which contribute to the distortion of the neurogenic niche in the hippocampus upon seizures that resemble to some extent the pathology of mesial temporal lobe epilepsy (MTLE). By applying in vitro assays, biochemistry and in vivo studies the authors identify the EGFR pathway and Zinc release as cause for reactive NSCs, the generation of reactive astrocytes and the granule cell dispersion in the DG. Importantly administration of the inhibitor gefitinib prevents numerous effects of kainic acid induced MTLE. This work is very well conducted and adds important new insight into the pathways that cause the detrimental effects on the adult neurogenic lineage that can further potentiate pathology in epilepsy. Prior to publication, it would be beneficial to clarify certain aspects in the manuscript.

We appreciate the positive feedback of the reviewer and their effort to provide detailed insight to improve the manuscript. We have addressed all the points.

Which cell type is upregulating EGFR expression in the hilus upon KA application (Fig. 1H) and could this cell type potentially add to the observed effects on proliferation and differentiation in the SGZ?

Because of their proximity to the SGZ, their co-expression of Nestin-GFP and their morphology the cells expressing EGFRs are more likely NSCs or ANPs. Might be some pericytes or oligodendrocyte progenitor cells could be labeled too but this is unlikely because the pericytes have a unique morphology that makes them readily distinguishable (Encinas et al. 2011).

It has been reported that EGFR is involved in microglia activation and the induction of reactive astrocytes as explained in the text. Therefore, even though the vast majority of EGFR+ cells are also Nestin-GFP+ cells, we cannot rule the possibility that some other cell types are involved and affect NSCs and neurogenesis secondarily. This notion is considered in the discussion.

Reference:

Encinas JM, Michurina TV, Peunova N, Park J-H, Tordo J, Peterson DA, Fishell G, Koulakov A, Enikolopov G. 2011. Division-coupled astrocytic differentiation and age-related depletion of neural stem cells in the adult hippocampus. *Cell Stem Cell* 8:566–579.

Does Gefitinib have a direct effect on the expression of EGFR? The expression seems to be increased upon treatment (Fig. 2A).

According to the literature Gefitinib targets the ATP-binding cleft epidermal growth factor receptor (EGFR) but no reports were found concerning a direct effect on its expression. Indirect effects cannot be ruled out and therefore we assessed the expression of EGFR in presence of Gefitinib in the in vitro model, finding no differences ($p=0.686$). We added this analysis to Figure 2B.

Does Gefitinib reverse or prevent GCD? Further I do not see evidence for GCD reverse in the referred Fig. 3A,B as this is the quantification of proliferation, nor in Fig. 3 D.

We meant to refer to “Supplementary Figure 3A and B” (and not Fig.3). Also, the quantification corresponds to the 14-day after KA time point. We thank the referees for catching this mistake.

Which cell type within the DG is supposed to release HB-EGF? Is there experimental indication of knowledge from the literature?

We thank the reviewer for bringing up this important point. According several reports, HB-EGF is widely distributed in neurons and neuroglia throughout the prenatal and postnatal brain of rats and can stimulate neurogenesis (Mishima et al., 1996; Hayase et al., 1998; Kulin J. et al., 2002; Nakagawa et al., 1998). In pathological conditions, HB-EGF mRNA and protein are induced in hippocampal CA3 and dentate gyrus (DG) neurons after global cerebral ischemia (Kawahara et al., 1999) and neonatal hypoxic–ischemic injury (Tanaka et al., 1999) in the rat.

References:

Hayase Y, Higashiyama S, Sasahara M, Amano S, Nakagawa T, Taniguchi N, Hazama F. Expression of heparin-binding epidermal growth factor-like growth factor in rat brain. *Brain Res.* 1998;784:163–178.

Kawahara N, Mishima K, Higashiyama S, Taniguchi N, Tamura A, Kirino T. The gene for heparin-binding epidermal growth factor-like growth factor is stress-inducible: its role in cerebral ischemia. *J Cereb Blood Flow Metab.* 1999;19:307–320.

Kunlin Jin, Xiao Ou Mao, Yunjuan Sun, Lin Xie, Lan Jin, Eiichiro Nishi, Michael Klagsbrun, David A Greenberg. Heparin-binding epidermal growth factor-like growth factor: hypoxia-inducible expression in vitro and stimulation of neurogenesis in vitro and in vivo. *J Neurosci.* 2002 Jul 1;22(13):5365-73. doi: 10.1523/JNEUROSCI.22-13-05365.2002.

Mishima K, Higashiyama S, Nagashima Y, Miyagi Y, Tamura A, Kawahara N, Taniguchi N, Asai A, Kuchino Y, Kirino T. Regional distribution of heparin-binding epidermal growth factor-like growth factor mRNA and protein in adult rat forebrain. *Neurosci Lett.* 1996;213:153–156.

Nakagawa T, Sasahara M, Hayase Y, Haneda M, Yasuda H, Kikkawa R, Higashiyama S, Hazama F. Neuronal and glial expression of heparin-binding EGF-like growth factor in central nervous system of prenatal and early-postnatal rat. *Brain Res Dev Brain Res.* 1998;108:263–272.

Tanaka N, Sasahara M, Ohno M, Higashiyama S, Hayase Y, Shimada M. Heparin-binding epidermal growth factor-like growth factor mRNA expression in neonatal rat brain with hypoxic/ischemic injury. *Brain Res.* 1999;827:130–138.

Why are no radial Nestin+ processes visible in Fig. 4A? Another field of view with a clearly detectable GFP+ process would be helpful for comparison.

For this technique we had to resort to epifluorescence microscopy in order to be able to capture the autometallography image. The treatment required for the tissue, together with the incompatibility to use immunostaining to enhance the GFP signal, make the thin radial process of NSCs almost undetectable. To demonstrate changes in Zn distribution we provide immunofluorescence against the zinc transporter “ZnT3” which takes up Zinc ions into synaptic vesicles showing an increase of ZnT3 after 3dPKA in agreement with Danscher data:

p14: "However, TPEN failed to reduce cell death, as a similar density of pycnotic nuclei was found in the GCL than KA+vehicle animals and significantly higher than saline+TPEN animals (Supplementary Figure 4F). These results suggest that despite mimicking the early effects of MTLE-HS on the neurogenic niche, endogenous zinc also exerts a neuroprotective effect."

If Zinc has a neuroprotective effect, wouldn't one expect to see higher levels of pyknotic nuclei in Fig. 4F KA+TPEN case?

This is an interesting question that requires better wording in the manuscript (see below). We meant that the lack of Zn due to chelation does not change cell death. Indeed, Zn can be both neuroprotective and neurotoxic (see review Choi S et al 2020). In our case, endogenous Zinc, released from mossy fibers in the hippocampus by seizure activity, is insufficient to alter the extent of neuronal death in agreement with previous work (Lees GC et al 1998).

"However, TPEN failed to alter cell death, as a similar density of pycnotic nuclei was found in the GCL than KA+vehicle animals and significantly higher than saline+TPEN animals (24864.24±630.84 pycnotic nuclei per mm³ for KA+TPEN respect to 2895.61±1635.08 pycnotic nuclei per mm³ for Saline+TPEN; Kruskal-Wallis p=0.014; Supplementary Figure 4F). These results suggest that despite mimicking the early effects of MTLE-HS on the neurogenic niche, endogenous zinc does not contribute to cell death."

References:

Choi S, Hong DK, Choi BY and Suh SW. Zinc in the Brain: Friend or Foe? *Int. J. Mol. Sci.* 2020, 21(23), 8941; <https://doi.org/10.3390/ijms21238941>

G J Lees, M P Cuajungco, W Leong. Effect of metal chelating agents on the direct and seizure-related neuronal death induced by zinc and kainic acid. *Brain Res.* 1998 Jul 13;799(1):108-17. doi: 10.1016/s0006-8993(98)00483-1.

In general, there are several wording, grammar and typ-o mistakes throughout the manuscript text which would need to be corrected.

We apologize for our errors. The manuscript has been thoroughly revised to correct all grammatical and typographic mistakes.

June 10, 2024

RE: Life Science Alliance Manuscript #LSA-2022-01840-TR

Prof. Juan Manuel Encinas-Pérez
Achucarro Basque Center for Neuroscience
Edif. Sede 3p. Campus UPV/EHU
Barrio Sarriena s/n
Leioa 48940
Spain

Dear Dr. Encinas-Pérez,

Thank you for submitting your revised manuscript entitled "HB-EGF activate EGFR to induce reactive neural stem cells in the mouse hippocampus after seizures". We would be happy to publish your paper in Life Science Alliance pending final revisions necessary to meet our formatting guidelines.

- please be sure that the authorship listing and order is correct
- please upload all figure files as individual ones, including the supplementary figure files; all figure legends should only appear in the main manuscript file
- please add ORCID ID for the corresponding (and secondary corresponding) author--you should have received instructions on how to do so
- please add the Twitter handle of your host institute/organization as well as your own or/and one of the authors in our system
- please consult our manuscript preparation guidelines <https://www.life-science-alliance.org/manuscript-prep> and make sure your manuscript sections are in the correct order
- please move your main, supplementary figure, and table legends to the main manuscript text after the references section
- please use the [10 author names et al.] format in your references (i.e., limit the author names to the first 10)
- please be sure that all authors are mentioned in the author contributions section in the manuscript file
- please remove the significance section after the abstract

A. FINAL FILES:

B. MANUSCRIPT ORGANIZATION AND FORMATTING:

Sincerely,

Reviewer #1 (Comments to the Authors (Required)):

The authors have addressed my initial concerns.

Reviewer #3 (Comments to the Authors (Required)):

In response to the reviewers points of critique and suggestions the manuscript "HB-EGF activate EGFR to induce reactive neural stem cells in the mouse hippocampus after seizures" has improved and is now ready for publication. All the points I raised have been sufficiently addressed and clarified. This work adds important insight into the pathophysiology of TLE in the hippocampus.

June 20, 2024

RE: Life Science Alliance Manuscript #LSA-2022-01840-TRR

Juan Manuel Encinas-Perez
Achucarro Basque Center for Neuroscience
Neuroscience
Campus UPV/EHU
Barrio Sarriena s/n
Leioa, Bizkaia 48940
Spain

Dear Dr. Encinas-Perez,

Thank you for submitting your Research Article entitled "HB-EGF activate EGFR to induce reactive neural stem cells in the mouse hippocampus after seizures". It is a pleasure to let you know that your manuscript is now accepted for publication in Life Science Alliance. Congratulations on this interesting work.

DISTRIBUTION OF MATERIALS:

Again, congratulations on a very nice paper. I hope you found the review process to be constructive and are pleased with how the manuscript was handled editorially. We look forward to future exciting submissions from your lab.

Sincerely,
